# Image Watermarks are Removable Using Controllable Regeneration from Clean Noise

**Yepeng Liu**[1], **Yiren Song**[2], **Hai Ci**[2], **Yu Zhang**[3], **Haofan Wang**[4],
**Mike Zheng Shou**[2], **Yuheng Bu**[1]

[1]University of Florida    [2]Show Lab, National University of Singapore
[3]Tongji University    [4]InstantX Team

`{yepeng.liu,buyuheng}@ufl.edu`, `{cihai03, mike.zheng.shou}@gmail.com`,
`yiren@nus.edu.sg`

## Abstract

Image watermark techniques provide an effective way to assert ownership, deter misuse, and trace content sources, which has become increasingly essential in the era of large generative models. A critical attribute of watermark techniques is their robustness against various manipulations. In this paper, we introduce a watermark removal approach capable of effectively nullifying state-of-the-art watermarking techniques. Our primary insight involves regenerating the watermarked image starting from a **clean Gaussian noise** via a controllable diffusion model, utilizing the extracted semantic and spatial features from the watermarked image. The semantic control adapter and the spatial control network are specifically trained to control the denoising process towards ensuring image quality and enhancing consistency between the cleaned image and the original watermarked image. To achieve a smooth trade-off between watermark removal performance and image consistency, we further propose an adjustable and controllable regeneration scheme. This scheme adds varying numbers of noise steps to the latent representation of the watermarked image, followed by a controlled denoising process starting from this noisy latent representation. As the number of noise steps increases, the latent representation progressively approaches clean Gaussian noise, facilitating the desired trade-off. We apply our watermark removal methods across various watermarking techniques, and the results demonstrate that our methods offer superior visual consistency/quality and enhanced watermark removal performance compared to existing regeneration approaches. Our code is available at `https://github.com/yepengliu/CtrlRegen`.

## 1 Introduction

As the large generative models continue to advance, the realism of AI-generated content has reached unprecedented levels. Those AI-generated contents are nearly indistinguishable from content created by humans. While this technological progress brings about excitement and efficiency, the difficulty of distinguishing AI-generated and human-made content can lead to severe problems, such as the spread of misinformation and the potential erosion of trust in digital media. In response, watermarking AI-generated content (Wen et al., 2024; Zhao et al., 2023a; Saberi et al., 2023; Zhao et al., 2023b; Lukas et al., 2023; Yang et al., 2024; Ci et al., 2024b;a; Kirchenbauer et al., 2023; Zhang et al., 2024a; Liu & Bu, 2024; Rezaei et al., 2024; He et al., 2024) has emerged as an effective solution, providing a proactive and reliable method to embed hidden information into AI-generated content for identification and traceability.

Recently, several promising watermarking methods for AI-generated images have emerged, typically embedding watermarks by perturbing the pixels or latent representations of the image. Low perturbation watermarks (Fernandez et al., 2023; Zhu, 2018; Fernandez et al., 2022; Zhang et al., 2019; Cox et al., 2007) lead to a small $\ell_2$ distance between watermarked and un-watermarked images in both pixel and latent space, making them potentially more vulnerable. On the other hand, watermark methods that induce high perturbation (Saberi et al., 2023; Zhao et al., 2023a) usually significantly modify the image or its representation, thereby resulting in enhanced robustness against

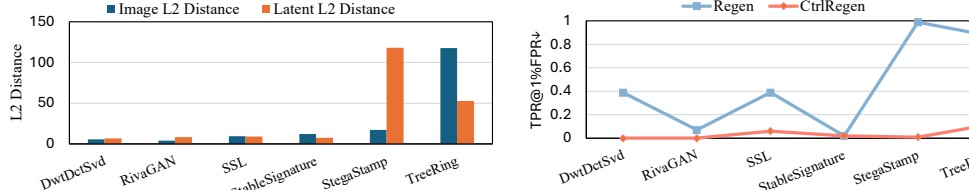

Figure 1: The left chart computes the $\ell_2$ distance between watermarked and un-watermarked images across different watermarking methods in both pixel and latent spaces, identifying StegaStamp and TreeRing as the high perturbation watermarks. The right chart shows the watermark removal performance of our method versus existing regeneration attacks across different watermarking techniques, highlighting the challenges of neutralizing high-perturbation watermarks with current methods.

various malicious manipulations, as shown in Figure 1. The robustness of image watermarks, being a crucial attribute, is targeted by numerous attacks aimed at removing the watermark.

For low perturbation watermarks, the regeneration attack (Zhao et al., 2023a; Saberi et al., 2023) is proposed to remove the invisible watermark that disturbs the image within a bounded $\ell_2$ distance range. Specifically, this method encodes the watermarked image from pixel to latent space, then uses the diffusion model's (Rombach et al., 2022) forward and reverse processes to add limited noise to the latent representation and denoise it to reconstruct the image. This method is effective for low-perturbation watermarks, which only cause limited pixel/representation perturbation and can thus be easily removed through limited noising and denoising steps. However, it struggles with high-perturbation watermarks like StegaStamp (Tancik et al., 2020) and TreeRing (Wen et al., 2024), which significantly alter the $\ell_2$ distance between watermarked and un-watermarked images in both pixel and latent spaces, as illustrated in Figure 1. The main reason is that the starting noise used for image reconstruction is derived by adding only a small amount of noise to the latent representation of the watermarked image. As a result, some hidden watermark information may still be retained and could be diffused into the regenerated image during the reverse process.

In terms of high perturbation watermarks, increasing the number of noising steps of regeneration or implementing the regeneration multiple times (An et al., 2024) can make the starting noise more closely resemble the pure Gaussian noise. This additional noise helps further disrupt the watermark structure, thereby enhancing the effectiveness of the watermark removal. However, as the number of noising steps increases, there is a significant sacrifice in image quality, as illustrated in Figure 5 and 7. This results in visible distortion and artifacts, and the image may even lose its semantic integrity.

The existing regeneration method struggles with balance: few noising steps are inadequate to eliminate high perturbation watermarks, while more noising steps significantly compromise the image quality and consistency between the watermarked and cleaned images. Therefore, it is a significant challenge to *implement a regeneration attack that effectively removes watermark structures for both low and high perturbation watermarks while maintaining image quality and consistency.*

In this paper, we introduce a novel watermark removal method, CtrlRegen, designed to remove image watermarks through a controllable regeneration process. Our core idea is to use **clean noise** sampled from the Gaussian distribution as a starting point for the denoising process of the diffusion model. This strategy ensures that watermark information is thoroughly cleaned up from both pixel and latent spaces. One significant challenge is controlling the diffusion model to generate the cleaned image from a clean Gaussian noise while maintaining image quality and content consistency. To address this, we train a plug-and-play semantic control adapter and spatial control network. These components extract semantic and spatial information from the watermarked image and use this information as conditions to guide the denoising process towards a high degree of consistency between the watermarked and cleaned images.

To offer a versatile solution that adapts to varying degrees of watermark robustness, we further propose an adjustable and controllable regeneration method, CtrlRegen+, as shown in Figure 2. This method begins by noising the latent representation of the image, and then controls the forward process starting from this noised latent representation. We can adjust the number of noising steps to regulate the degree of watermark destruction. With an increased number of noising steps, the latent representation becomes closer to pure noise, resulting in a more thorough destruction of the watermark information. It is worth noting that, unlike the uncontrolled regeneration method, our

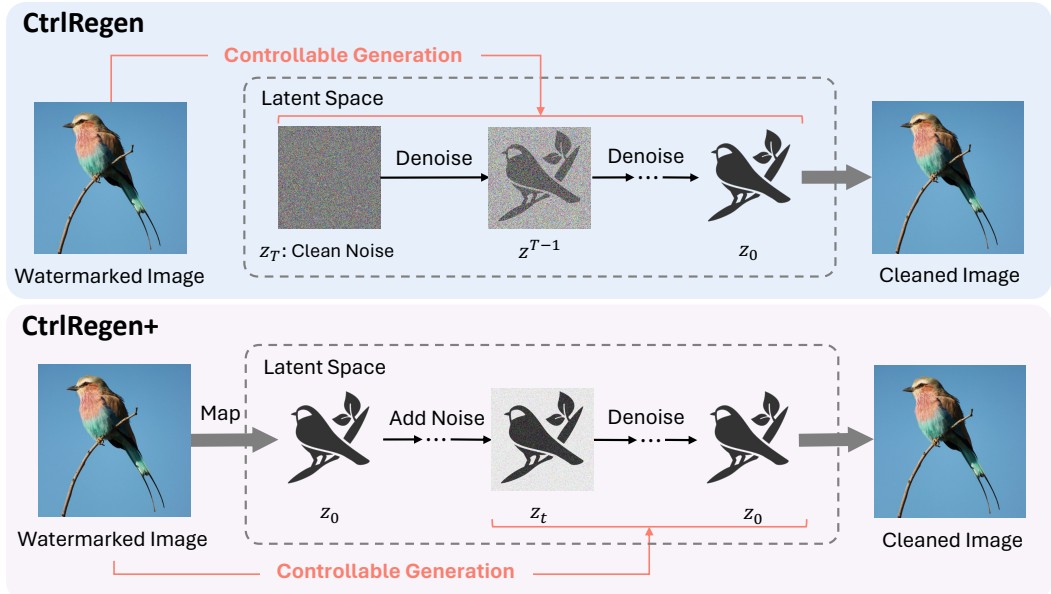

Figure 2: Overview of the proposed method. CtrlRegen controls the regeneration of watermarked images from a clean noise without any watermark information. CtrlRegen+ first encodes the watermarked image into a latent representation and introduces varying levels of noise based on the robustness of the watermark. It then controls the denoising process to reconstruct the image.

controllable method still preserves high image quality and visual similarity with the watermarked image, even with a high number of noising steps, as presented in Figure 5.

We conduct experiments to implement our watermark removal methods across various watermark methods, including low and high perturbation watermarks. The results demonstrate that our Ctrl-Regen effectively reduces the detection performance (TPR@1%FPR) of StegaStamp from 1.00 to 0.01 and of TreeRing from 0.99 to 0.12. Conversely, the uncontrolled regeneration method proves less effective for these two watermark methods. Moreover, our CtrlRegen+ achieves better image quality/consistency while maintaining the same watermark removal performance compared to the uncontrolled regeneration approach.

## 2 RELATED WORK

### 2.1 IMAGE WATERMARK METHODS

Watermarking techniques offer an active and reliable method to trace the source of images or protect copyright. Traditional methods embed watermark information directly into the generated images (Rouhani et al., 2018; Chen et al., 2019; Jia et al., 2021). With the development of large generative models, such as Stable Diffusion (Rombach et al., 2022), watermark information can now also be integrated seamlessly into the process of creating digital content. We broadly categorize watermarking methods into post-hoc methods and in-generation methods based on the stage at which the watermark is embedded. Post-hoc methods embed watermarks into a given image using techniques such as encoder-decoder (e.g., HiDDeN (Zhu, 2018), Stegastamp (Tancik et al., 2020)), optimization (e.g., SSL (Fernandez et al., 2022)), or wavelet transforms (e.g., DwtDctSvd (Cox et al., 2007)). In contrast, in-generation methods introduce watermarks during image generation by modifying components of the generative model, such as initial noise (e.g., Tree-Ring (Wen et al., 2024), RingID (Ci et al., 2024b)) or the VAE decoder (e.g., StableSignature (Fernandez et al., 2023), WMAdapter (Ci et al., 2024a)). We evaluated the effectiveness of our proposed methods on both post-hoc and in-generation watermarking methods.

### 2.2 IMAGE WATERMARK REMOVING METHODS

Robustness, a crucial attribute of image watermarking, is assessed through a range of watermark removal attacks (Hu et al., 2024; Kassis & Hengartner, 2024), including editing attacks, regenera-

tion attacks, and adversarial attacks. The editing attack comprises typical image manipulations such as cropping, compression, rotation, brightness/contrast adjustments, and the addition of Gaussian noise. These manipulations simulate common alterations that images might undergo in practice. Most existing watermarking methods demonstrate robustness against these basic operations. Recently, regeneration attack (Zhao et al., 2023a; Saberi et al., 2023), as a no-box attack, proposes to remove the image watermark through the noising and denoising process of pre-trained diffusion model or encoding and decoding process of variational autoencoder (VAE). The regeneration attack demonstrates a powerful ability to remove watermarks for methods that cause minimal perturbation to the pixel or latent space of a watermarked image. However, it tends to be less effective under limited noising steps for watermarking methods that induce high perturbation. For those high perturbation watermarking methods, the adversarial attacks (Saberi et al., 2023; Lukas et al., 2023; Jiang et al., 2023) provide an effective method to generate adversarial images that evade watermark detection. These methods treat the watermark detector as a classifier and introduce adversarial perturbations to the watermarked image through optimization to deceive the detector. However, this strategy demands more unrealistic capabilities from attackers, including knowledge of the watermarking method (Lukas et al., 2023), access to the watermarked large generative models in the white-box setting (Saberi et al., 2023), or the ability to make multiple uninterrupted queries to the API of watermark detector (Jiang et al., 2023). Moreover, these methods are image-specific and watermark-specific, which means that attackers need to tailor the perturbation for each image according to different watermarking methods. This process is both time-consuming and computationally intensive. In this paper, we focus on the regeneration attack and propose a controllable regeneration attack combined with our proposed control techniques.

## 2.3 DIFFUSION MODELS

Diffusion probability models (Song et al., 2020; Ho et al., 2020) are advanced generative models that restore original data from pure Gaussian noise by learning the distribution of noisy data at various levels of noise. With their powerful capability to adapt to complex data distributions, diffusion models have achieved outstanding achievements in several domains, including image synthesis (Rombach et al., 2022; Peebles & Xie, 2023), image editing (Brooks et al., 2023; Hertz et al., 2022; Zhang et al., 2024c;d), and even 3D content creation (Poole et al., 2022). Among them, Stable Diffusion (Rombach et al., 2022) (SD), a notable example, employs a UNet architecture and iteratively generates images with impressive text-to-image capabilities through extensive training on large-scale text-image datasets. Alongside these developments, controllable image generation has seen enhancements from methods like ControlNet (Zhang & Agrawala, 2023) and T2I-adapter (Mou et al., 2023), which utilize multimodal inputs such as depth maps and segmentation maps to significantly increase the controllability over the generated images. Furthermore, subject-driven image generation techniques now range from those requiring test-time fine-tuning (Gal et al., 2022; Ruiz et al., 2022; Kumari et al., 2023; Hu et al., 2022) to those operating entirely fine-tuning-free (Ye et al., 2023; Zhang et al., 2024b), each offering varying degrees of adaptability and computational demand. In this paper, we propose an image watermarking removal algorithm for watermark removal based on ControlNet Zhang & Agrawala (2023) and IP-Adapter Ye et al. (2023).

## 3 THE PROPOSED CONTROLLABLE REGENERATION ATTACK

### 3.1 OVERVIEW

The core idea behind our proposed method is to regenerate the watermarked image starting from clean noise. A controllable diffusion model is then designed to maintain the consistency between the watermarked image and the cleaned image during the denoising process, with the watermarked image serving as a conditional input.

The workflow of our method is presented as Figure 3. Specifically, given a watermarked image $x_w \in \mathbb{R}^N$, starting latent representation $z$, and the generation function $\mathcal{G} : \mathbb{R}^n \times \mathbb{R}^N \to \mathbb{R}^N$, we can obtain the cleaned image:

$$\tilde{x} = \mathcal{G}(z, x_w, \hat{x}_w), \tag{1}$$

where $\hat{x}_w = f(x_w) \in \mathbb{R}^N$ represents the edge-detected image obtained from $x$ using edge detection algorithm. For CtrlRegen, input $z$ is $\epsilon \sim \mathcal{N}(0, I_n)$. For CtrlRegen+, input $z$ is given by $z_{t^*} =$

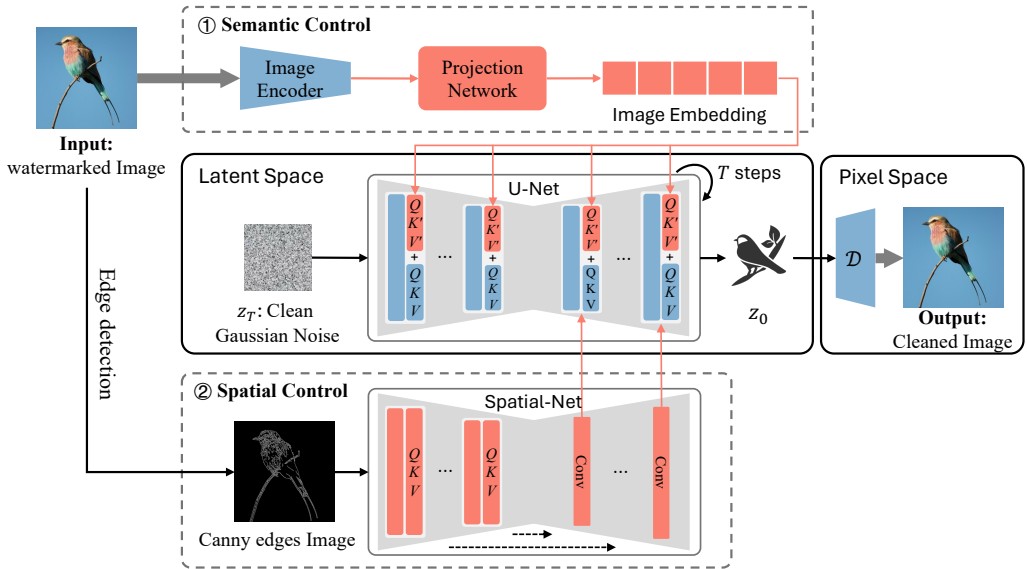

Figure 3: Workflow of the controllable regeneration. The red modules represent the trainable parameters, while the blue modules represent the pre-trained and fixed parameters. Semantic control is applied through inserted cross-attention modules using extracted image embedding as a condition. Spatial control is achieved by integrating the outputs from the convolution layers of Spatial-Net into the decoder blocks of the U-Net.

$\sqrt{\alpha_{t^*}} z_0 + \sqrt{1 - \alpha_{t^*}} \epsilon$, where $t^* \in [0, T]$ is the number of noising steps, $z_0 = \mathcal{E}(x_w)$ is the latent representation obtained by encoding the watermarked image using an encoder, and $\alpha_{t^*}$ represents the variance schedule of the noise $\epsilon \sim \mathcal{N}(0, I_n)$, which gradually decreases from 1 to 0 as $t^*$ increases from 0 to $T$.

Compared to the previous uncontrolled regeneration methods, one important goal of our approach is to control the generation process starting from $z$ to maintain the consistency between $x_w$ and $\tilde{x}$. To achieve this, we propose two methods designed to control the semantic content and spatial distribution of the images, respectively.

## 3.2 REGENERATION CONTROL MODEL TRAINING

### 3.2.1 SEMANTIC CONTROL

One key challenge in semantic control is preserving the semantic content of watermarked images while effectively destroying the watermark information. Previous uncontrolled regeneration methods achieved this by initially destructing the $z_0$ with random Gaussian noise via the forward process of the diffusion model, followed by reconstructing the image through an uncontrolled reverse process. While this method of destructing through a limited number of noising and denoising steps proves effective for removing low perturbation watermarks, it falls short in eliminating watermark that causes high perturbation in the latent space. Drawing inspiration from SD model, which employs a text encoder to convert text prompts into text embeddings and then using the cross-attention mechanism (Vaswani, 2017) to ensure the generated image semantically aligns with the text, we compress the watermarked image into an image embedding that preserves only semantic content. This embedding is then used to control the generation process via cross-attention, ensuring the regenerated image retains semantic accuracy without the watermark information.

We train a semantic control adapter to facilitate semantic control as depicted in the upper branch of Figure 3, which includes an image encoder, projection network, and newly implemented decoupled cross-attention layer. Our training approach for the projection network and cross-attention layer draws on strategies similar to those used in the IP-Adapter (Ye et al., 2023). Specifically, we employ pre-trained DINOv2 (Oquab et al., 2023) as our image encoder to extract the image feature from the watermarked image, capitalizing on its high-performance capability to extract rich visual features.

A trainable projection network is then used to transform the extracted image feature into image embedding, which is used to control the generation through cross-attention in the following step.

To preserve the original SD model and control generation using image embedding as a condition, an additional trainable image cross-attention layer is integrated into the U-Net within the SD model. This layer is designed to compute the attention using the image embedding $\phi(x)$, such that

$$\texttt{Attention}(Q, K', V') = \texttt{softmax}\left(\frac{Q(K')^T}{\sqrt{d}}\right) \cdot V', \tag{2}$$

where $Q = W_Q \cdot \varphi(z^t)$, $K' = W'_K \cdot \phi(x)$, and $V' = W'_V \cdot \phi(x)$. Here, the $\varphi(z^t)$ is the intermediate representation within the U-Net. $W'_K$ and $W'_V$ are learnable matrices specifically for image cross-attention. $W_Q$ is the same projection matrix employed in the text cross-attention. During the training and inference process, text cross-attention and image cross-attention operate concurrently. The combined attention mechanism is expressed as $Z = \texttt{Attention}(Q, K, V) + \texttt{Attention}(Q, K', V')$, where $K$ and $V$ are the key and value matrices associated with text cross-attention, and $K'$ and $V'$ pertain to the image cross-attention. Specifically, we set the text prompt to be empty during both the training and inference process. Therefore, only the image embedding influences and controls the generation process.

During the training process, the parameters of the image encoder $\theta_e$ and the SD $\theta_d$ are kept frozen, while only the projection network $\theta_p$ and the image cross-attention layer $\theta_a$ are trained. Given the training image dataset $X$, the training objective can be expressed as:

$$\mathcal{L}_{\theta_p, \theta_a} = \mathbb{E}_{x, \epsilon \sim \mathcal{N}(0, I_n), t}[\|\epsilon - \epsilon_{\theta_S}(z_t, \phi_{\theta_e, \theta_p}(x), t)\|_2^2], \tag{3}$$

where $x \in X$ is the input image, $\phi_{\theta_e, \theta_p}(x)$ maps the input image to an embedding. For $t = 1, \cdots, T$, $z_t$ is the latent representation at different time steps, $\epsilon_{\theta_S}(z_t, \cdot, t)$ represents a sequence of U-Net used to predict the noise given the input $z_t$, $t$ and condition $\phi_{\theta_e, \theta_p}(x)$, where $\theta_S$ includes fixed $\theta_d$ and $\theta_e$, and trainable $\theta_p$ and $\theta_a$. We use only the image part of the dataset as the generation condition.

### 3.2.2 SPATIAL DISTRIBUTION CONTROL

The semantic control adapter facilitates a coarse-grained semantic content alignment between the watermarked image and the regenerated image, ensuring that most of the semantic content is preserved. However, it struggles to control the finer details and layout during the generation process. To address this, enhancing spatial control of regeneration is another key challenge. Incorporating an edge-detected image as an additional condition offers an effective solution, as it provides crucial spatial information without introducing extraneous watermark information. To better leverage spatial information from the edge-detected image, we propose to use a spatial control network. This network is designed to extract spatial features, which are then integrated into the denoising process of U-Net, thus enhancing spatial distribution in the regenerated image.

An Edge-detected image emphasizes the boundaries and contours within an image by identifying the rapid changes in intensity, which correspond to object edges. Typically rendered as a binary image, it features edges marked in white against non-edge areas in black. Specifically, we use Canny edge images extracted from the watermarked image using the Canny detection method (Canny, 1986). In our method, we adopt the ControlNet (Zhang & Agrawala, 2023) structure for our spatial control network, as shown in the lower branch of Figure 3. After applying the spatial control network to U-Net, the output of the neural blocks within U-Net is expressed as:

$$\zeta(z_t^i, \phi_{\theta_e, \theta_p}(x), \hat{x}, t) = \mathcal{F}_{\theta_u}(z_t^i, \phi_{\theta_e, \theta_p}(x), t) + \mathcal{H}_{\theta_C}(\hat{x}^i, t), \tag{4}$$

where $z_t^i$ is the intermediate representation between different neural blocks within U-Net, $x$ is the original image, $\hat{x} = f(x)$ represents the edge-detected image, $\mathcal{F}_{\theta_u}(\cdot)$ is the original neural block of U-Net, $\mathcal{H}_{\theta_C}(\cdot)$ represents the neural blocks and convolution layers of spatial network, $\hat{x}^i$ refers to the intermediate representation within the spatial control network that corresponds dimensionally with $z_t^i$, in which $\hat{x}^0$ is derived from $\hat{x}$ through an encoder and a convolution layer.

To enhance compatibility and integration among the components, we combine the semantic control adapter, spatial control network, and SD within a unified framework. We fix the parameters of the already trained semantic control adapter and the SD and tune the spatial control network $\theta_C$ to optimize its performance in conjunction with the other components. Now, with both image embedding

and edge-detected image as conditions, the training objective is:

$$\mathcal{L}_{\theta_C} = \mathbb{E}_{x,\hat{x},\epsilon \sim \mathcal{N}(0,I_n),t}[\|\epsilon - \epsilon_\theta(z_t, \phi_{\theta_e,\theta_p}(x), \hat{x}, t)\|_2^2], \tag{5}$$

where $\epsilon_\theta(z_t, \phi_{\theta_e,\theta_p}(x), \hat{x}, t)$ is the U-Net with $z_t$ and $t$ as input and $\phi_{\theta_e,\theta_p}(x)$ and $\hat{x}$ as condition, $\theta$ includes both $\theta_S$ and $\theta_C$.

### 3.3 REGENERATION WITH CONTROL

Once our semantic control adapter and spatial control network are fully trained, they are ready to be deployed for direct watermark removal, requiring only a single watermarked image. The entire inference process of CtrlRegen is outlined in Algorithm 1 of Appendix B. We sample the initial noise from a pure Gaussian distribution. Then the extracted image embedding and edge-detected image serve as conditions, providing the necessary semantic and spatial information. Two well-trained networks are integrated with the backbone SD to denoise across $T$ timesteps. Finally, the cleaned image is obtained by decoding the denoised latent representation.

The inference process of CtrlRegen+ is detailed in Algorithm 2 of Appendix B. Unlike CtrlRegen, which utilizes pure Gaussian noise as a starting point, CtrlRegen+ first adds noise to the latent representation and then uses this noised representation as a starting point to reconstruct the image. The noising step can be selected based on the strength of different watermark methods.

## 4 EXPERIMENTS

### 4.1 EXPERIMENT SETTING

**Datasets.** We evaluate our watermark removal performance using two datasets. For the post-hoc watermarking methods, we sample 1000 real photos from the MIRFLICKR (Huiskes & Lew, 2008), which consists of a comprehensive image collection sourced from the social photography platform Flickr. For the in-generation watermarking methods, we sample 1000 prompts from a large-scale text-to-image prompt dataset, DiffusionDB (Wang et al., 2022), to generate watermarked images using generative models. Additionally, we train the semantic control adapter using 10 million images sampled from LAION-2B (Schuhmann et al., 2022) and COYO-700M [1]. The spatial control network is trained using 118k image-canny pairs from MSCOCO (Lin et al., 2014).

**Image Watermark Methods.** To demonstrate the effectiveness of our watermark removal method, we evaluate it against seven diverse watermarking methods, including TreeRing (Wen et al., 2024), StableSignature (Fernandez et al., 2023), StegaStamp (Tancik et al., 2020), HiDDeN (Zhu, 2018), SSL (Fernandez et al., 2022), RivaGAN (Zhang et al., 2019) and DwtDctSvd (Cox et al., 2007). These methods encompass both low and high-perturbation watermarks, providing a comprehensive evaluation of our approach's capabilities. For multi-bit watermarking methods, the specific number of bits used for each method is provided in Table 4 of Appendix A.

**Image Watermark Removal Baselines.** Our method is compared with two regeneration methods Regen (Zhao et al., 2023a) and Rinse (An et al., 2024). Regen employs the diffusion model to regenerate watermarked images through a process of noising and denoising. Rinse iteratively applies the Regen method multiple times to improve the efficacy of watermark removal. For the experimental results shown in Table 1, we set the noising and denoising steps to 70 for Regen, while Rinse applies the Regen process twice.

**Implementation Details.** We employ Stable Diffusion-v1.5 (Rombach et al., 2022) as the backbone for our model, maintaining its parameters in a frozen state to preserve the original capabilities. For the semantic control adapter, we integrate DINOv2-giant (Oquab et al., 2023) as the image encoder, also keeping its parameters frozen to leverage its pre-trained strengths. The training of the semantic control adapter is conducted on 8 NVIDIA A100 GPUs, and the batch size is set to 8 per GPU. The training of the spatial control network is carried out on 8 NVIDIA A100 GPUs with a batch size of 4 per GPU. At the inference stage, we conduct experiments on a single NVIDIA RTX 4090.

**Evaluation Metrics.** The watermark removal performance is evaluated using the TPR@1%FPR, which calculates the True Positive Rate (TPR) when the False Positive Rate (FPR) is constrained

---

[1]https://github.com/kakaobrain/coyo-dataset

to 1%. It is crucial to ensure a low rate of misclassification of the unwatermarked image as watermarked. Moreover, we calculate the average bit accuracy for multi-bit watermark methods, which measures the percentage of correctly recovered watermark bits. For better comparison, the average bit accuracy and TPR@1%FPR before (Avg Bit Acc B and T@1%F B) and after (BitAcc A and T@1%F A) the attack are both presented.

To evaluate visual similarity and quality, we employ two types of metrics: full-reference and non-reference assessment. For full-reference assessment, we use CLIP-FID (Kynkäänniemi et al., 2022) and Peak Signal-to-Noise Ratio (PSNR) to evaluate the visual similarity between watermarked images and regenerated images. For non-reference assessment, we adopt two state-of-art methods, Q-Align (Wu et al., 2023) and LIQE Zhang et al. (2023), to evaluate the quality of regenerated images. Q-align uses large multi-modality models to evaluate image quality by aligning machine-generated scores with human judgment by leveraging discrete text-defined levels for scoring. The LIQE uses a multitask learning approach, combining scene classification and distortion identification with CLIP-derived embeddings, to assess image quality in a non-reference manner. These metrics allow us to comprehensively analyze the image quality both with and without a reference image, ensuring a thorough assessment of the visual outcomes.

Table 1: Comprehensive Performance of attacks between our CtrlRegen and other methods.

| Watermarks | Attacks | BitAcc B ↑ | BitAcc A ↓ | T@1%F B ↑ | T@1%F A ↓ | CLIP-FID ↓ | PSNR ↑ | Q-Align ↑ | LIQE ↑ |
|---|---|---|---|---|---|---|---|---|---|
| DwtDctSvd | Regen | 1.00 | 0.64 | 1.00 | 0.39 | 8.91 | 26.01 | 3.34 | 2.82 |
| | Rinse | 1.00 | 0.53 | 1.00 | 0.11 | 11.50 | 23.83 | 2.95 | 2.27 |
| | **CtrlRegen** | 1.00 | 0.46 | 1.00 | 0.00 | 8.68 | 19.13 | 3.63 | 3.76 |
| RivaGAN | Regen | 1.00 | 0.55 | 1.00 | 0.07 | 4.44 | 25.93 | 3.26 | 2.68 |
| | Rinse | 1.00 | 0.50 | 1.00 | 0.02 | 7.39 | 23.72 | 2.87 | 2.11 |
| | **CtrlRegen** | 1.00 | 0.48 | 1.00 | 0.00 | 4.24 | 19.53 | 3.62 | 3.69 |
| SSL | Regen | 0.99 | 0.68 | 1.00 | 0.39 | 6.06 | 22.25 | 2.66 | 2.54 |
| | Rinse | 0.99 | 0.59 | 1.00 | 0.10 | 8.89 | 20.34 | 2.28 | 1.93 |
| | **CtrlRegen** | 0.99 | 0.56 | 1.00 | 0.06 | 5.64 | 19.07 | 3.22 | 3.15 |
| StableSignature | Regen | 0.99 | 0.49 | 1.00 | 0.02 | 1.91 | 24.16 | 3.86 | 3.67 |
| | Rinse | 0.99 | 0.47 | 1.00 | 0.10 | 4.15 | 21.85 | 3.50 | 2.98 |
| | **CtrlRegen** | 0.99 | 0.49 | 1.00 | 0.02 | 1.83 | 19.03 | 3.97 | 4.02 |
| StegaStamp | Regen | 1.00 | 0.88 | 1.00 | 0.99 | 6.48 | 22.34 | 3.06 | 3.53 |
| | Rinse | 1.00 | 0.77 | 1.00 | 0.94 | 10.73 | 21.31 | 2.67 | 2.71 |
| | **CtrlRegen** | 1.00 | 0.49 | 1.00 | 0.01 | 5.27 | 19.10 | 3.62 | 3.77 |
| TreeRing | Regen | — | — | 0.99 | 0.87 | 2.84 | 25.59 | 4.03 | 3.96 |
| | Rinse | — | — | 0.99 | 0.61 | 5.83 | 23.28 | 3.69 | 3.29 |
| | **CtrlRegen** | — | — | 0.99 | 0.12 | 1.63 | 19.32 | 4.17 | 4.34 |

## 4.2 MAIN RESULTS

### 4.2.1 CTRLREGEN

**Watermark Removal Performance of CtrlRegen.** Table 1 presents the watermark detection performance across various watermarking methods, both before and after the application of different watermark removal attacks. It shows that all watermarking methods have great detection performance before attacks. For low perturbation watermarking methods such as DwtDctSvd, RivaGAN, SSL and StableSignature, the Regen and Rinse exhibit effective watermark removal performance. However, for high perturbation methods like StegaStamp and TreeRing, Regen and Rinse become ineffective. This is because StegaStamp and TreeRing induce significant disturbances in both pixel-space and latent-space. Specifically, Regen's noising and denoising process is applied to the latent representation of the watermarked image, and the limited number of noising steps is insufficient to completely disrupt the watermark structure embedded within the latent representation. This limitation hinders its ability to effectively manage substantial perturbations in the latent space. Rinse extends the approach by running Regen multiple times in an attempt to further disrupt the watermark structure. Despite the enhancement, the effectiveness is still limited, especially for StegaStamp, which still has 0.94 TPR@1%FPR. Our method regenerates the watermarked image by starting from a clean Gaussian noise in the latent space, which inherently contains no watermark structure within the latent representation. Therefore, our watermark removal approach achieves notable performance, reaching 0.01 and 0.12 TPR@1%FPR for StegaStamp and TreeRing, respectively.

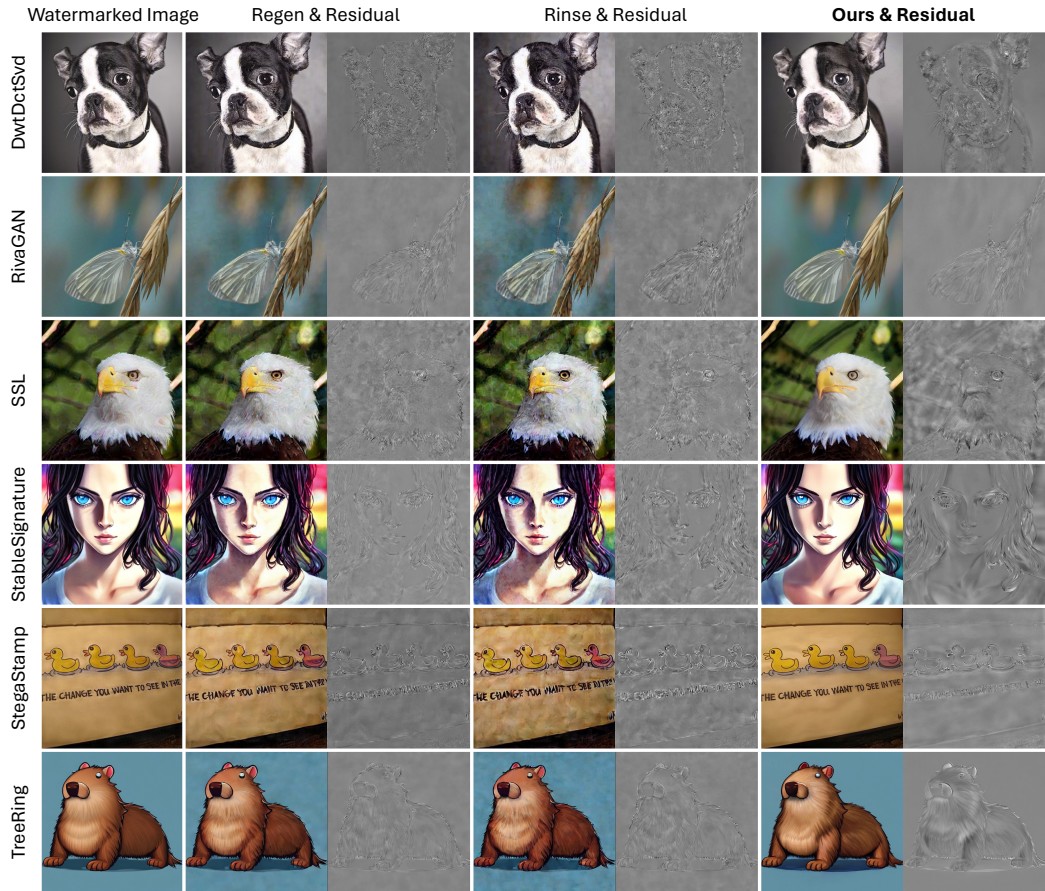

Figure 4: Examples of different watermark removal attacks on different watermarking methods.

**Visual Similarity and Quality of CtrlRegen.** The regenerated images are assessed from two key aspects: visual similarity and quality. Table 1 presents these measurements for various watermark removal methods. For visual similarity, our CtrlRegen achieves a lower CLIP-FID score compared to baselines. This lower score suggests that the distribution of our regenerated images more closely approximates that of the original watermarked images, reflecting better preservation of visual characteristics and overall image integrity. CtrlRegen exhibits a lower score for pixel-level measurement, PSNR, compared to baselines. However, it does not necessarily indicate severe degradation in similarity. The changes that lead to a lower PSNR can actually harmonize well with the content of the image without making the regenerated image appear unnatural. On the contrary, although the Regen and Rinse may achieve higher PSNR scores, it results in visible artifacts and degradation in the regenerated images compared to the original watermarked images, as illustrated in Figure 4. We provide more discussion and examples (Figure 8) for this problem in Appendix A. To further substantiate that images regenerated by our method exhibit superior image quality, we employ two image quality measurements: Q-Align and LIQE. These metrics are used to quantitatively assess and compare the quality of the regenerated images. From Table 1, our method achieves better image quality compared to Regen and Rinse.

### 4.2.2 CTRLREGEN+

We evaluate our CtrlRegen+ against Regen across various aspects, including watermark removal effectiveness, visual resemblance, and image quality, by varying the number of noising steps. For high-perturbation watermarks, such as StegaStamp and TreeRing, we set the noising steps to be $\{100, 200, 300, 400, 500, 1000\}$ and sample from pure Gaussian noise. We do not set a noising step number larger than 500 for Regen, as at this number of noising steps, the noised latent representation approaches pure Gaussian noise. Consequently, the uncontrolled regeneration process would likely produce an image completely unrelated to the original watermarked image.

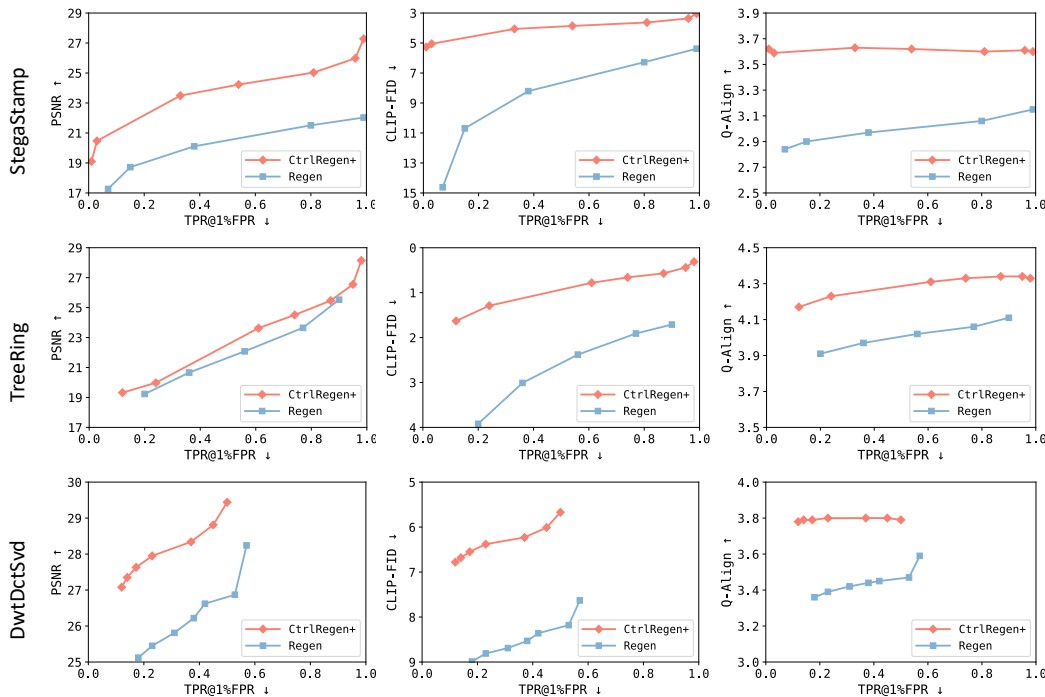

Figure 5: Performance of CtrlRegen+ compared to Regen with a varying number of noising steps on high and low perturbation watermarks, respectively. We invert the CLIP-FID score to ensure that the top-left represents better performance across all figures.

The first two rows of Figure 5 illustrate the relationship between watermark removal performance and visual similarity/quality across different noising steps for high perturbation watermarks. Compared to Regen, CtrlRegen+ not only maintains high image quality and similarity but also achieves superior watermark removal performance. Specifically, for Regen, using a high number of noising steps can indeed remove the watermark but also lead to a significant degradation in the quality of images, as shown in Figure 7 of Appendix A. It will make the watermark removal meaningless.

For low-perturbation watermarks, a small number of noising steps is sufficient to remove the watermark. Therefore, we evaluate the performance of CtrlRegen+ using a range of relatively low noising steps, i.e., $\{20, 40, 60, 80, 100, 120, 140\}$. As shown in the third row of Figure 5, even with fewer noising steps, our CtrlRegen+ method still demonstrates superior image quality and similarity compared to the Regen approach while achieving comparable watermark removal performance. More results of CtrlRegen+ on low perturbation watermarks are presented in Figure 6 of Appendix A.

Furthermore, we conduct an ablation study detailed in Appendix A to investigate the impact of semantic and spatial control on consistency.

## 5 CONCLUSION

In this paper, we introduce a controllable regeneration attack that, combined with proposed control techniques, effectively removes image watermarks by thoroughly eliminating the watermark information in the latent space. Our experiments demonstrate improved visual consistency and image quality compared to existing uncontrolled regeneration attacks under the same removal performance. We note that our attack is a no-box approach, meaning that the attacker only needs one watermarked image to remove watermarks without requiring knowledge of the watermarking scheme, detection rules, or access to the detector. By demonstrating the ability to defeat robust watermarking techniques, we highlight the urgent need to develop stronger watermarking solutions that can withstand this attack. In this work, the consistency of regeneration may be constrained to a certain extent by the backbone model we employed. In future work, we will explore more advanced backbone models, such as SD-v3 (Esser et al., 2024), and refine our control techniques for further improvement.

ETHICS STATEMENT

Our primary motivation for developing a more effective watermark removal method is to uncover the vulnerabilities in current watermarking techniques. By demonstrating the weaknesses in existing watermarking methods, we aim to underscore the urgent need for more robust and resilient watermarking solutions. Additionally, **our methods will serve as a benchmark for assessing the robustness of future watermarking techniques and fostering their advancement.** The proposed watermark removal techniques may pose ethical risks, including potential copyright infringement, undermining digital rights management, and enabling the unauthorized use of protected content. It is essential that **our methods should be used responsibly and in compliance with local regulations, ensuring this work contributes to strengthening, rather than compromising, digital watermarking security.**

ACKNOWLEDGEMENT

The UF team acknowledges UFIT Research Computing for providing computational resources and support that have contributed to the research results reported in this publication. The NUS team is only supported by the Ministry of Education, Singapore, under the Academic Research Fund Tier 1 (FY2023).

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

## A    ADDITIONAL EXPERIMENTAL RESULTS AND ABLATION STUDY

**CtrlRegen+ for Low perturbation watermarks.** Figure 6 displays the results of CtrlRegen+ compared to Regen across three low-perturbation watermarks: RivaGAN, SSL, and StableSignature. For SSL, the number of noising steps is set to $\{20, 60, 100, 140, 180, 220, 260, 300, 340\}$. From the figure, it is evident that SSL remains relatively robust compared to other low perturbation watermarks when number of noising steps is small. Our CtrlRegen+ presents better consistency compared to Regen at the same TPR@1%FPR. Additionally, the quality of images regenerated by Regen deteriorates with an increase in noising steps, whereas the quality of images from CtrlRegen+ improves. This improvement occurs because SSL introduces visible perturbations to the image, which are gradually removed as the number of noising steps increases in the CtrlRegen+ process. For RivaGAN and StableSignature, the number of noising steps is set to $\{20, 40, 60, 80, 100, 120, 140\}$. StableSignature is the most vulnerable to regeneration attacks, with even just 20 noising steps being sufficient to remove its watermark.

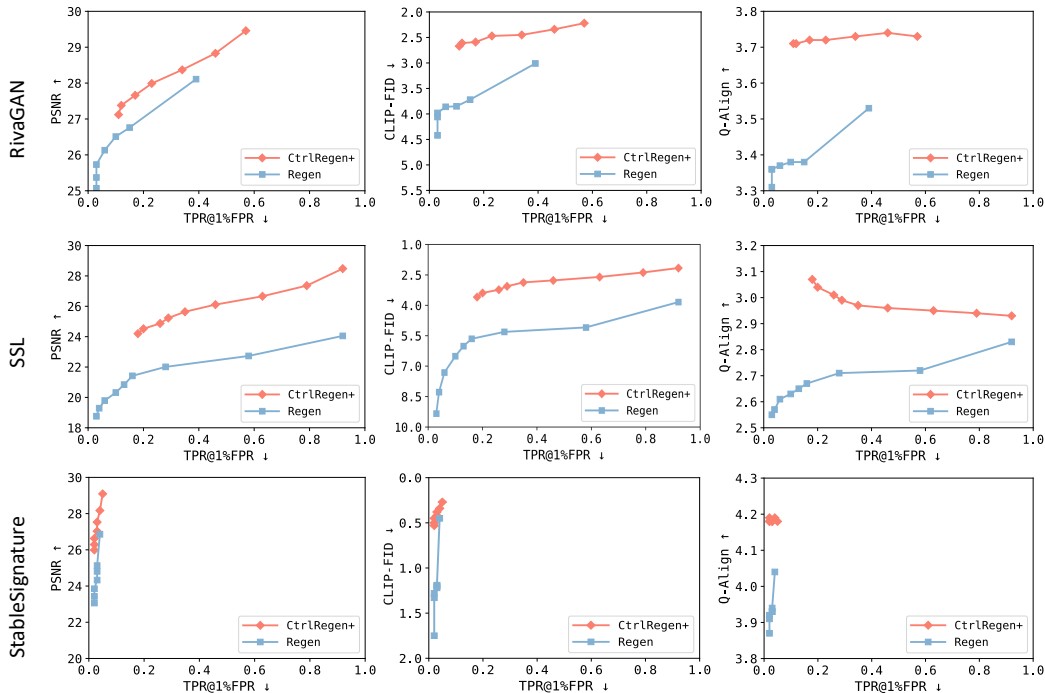

Figure 6: Performance of CtrlRegen+ compared to Regen on the remaining low perturbation watermarks. The noising step number is set to $\{20, 60, 100, 140, 180, 220, 260, 300, 340\}$ for SSL and $\{20, 40, 60, 80, 100, 120, 140\}$ for RivaGAN and StableSignature.

**Examples of CtrlRegen+.** Figure 7 shows the image regenerated by CtrlRegen+ and Regen at various noising steps. From those examples, we can observe that, as the number of noising steps increases, images regenerated by Regen become noisy and distorted. Conversely, images regenerated by CtrlRegen+ maintain high visual consistency and quality.

**Semantic Control and Spatial Control.** Our controllable regeneration process comprises two crucial components: semantic control and spatial control. The semantic control adapter extracts the semantics from the watermarked image, ensuring the basic semantic integrity of the regenerated image, though it may lack detailed accuracy. The spatial control network, on the other hand, manages more detailed aspects of the regeneration process. To demonstrate this, we regenerate the watermarked image using only semantic control and then again using both semantic and spatial controls together, allowing us to observe the enhancements in visual similarity and quality. Table 2 displays the corresponding results. It demonstrates that the usage of spatial control not only enhances the consistency of the image but also improves its quality.

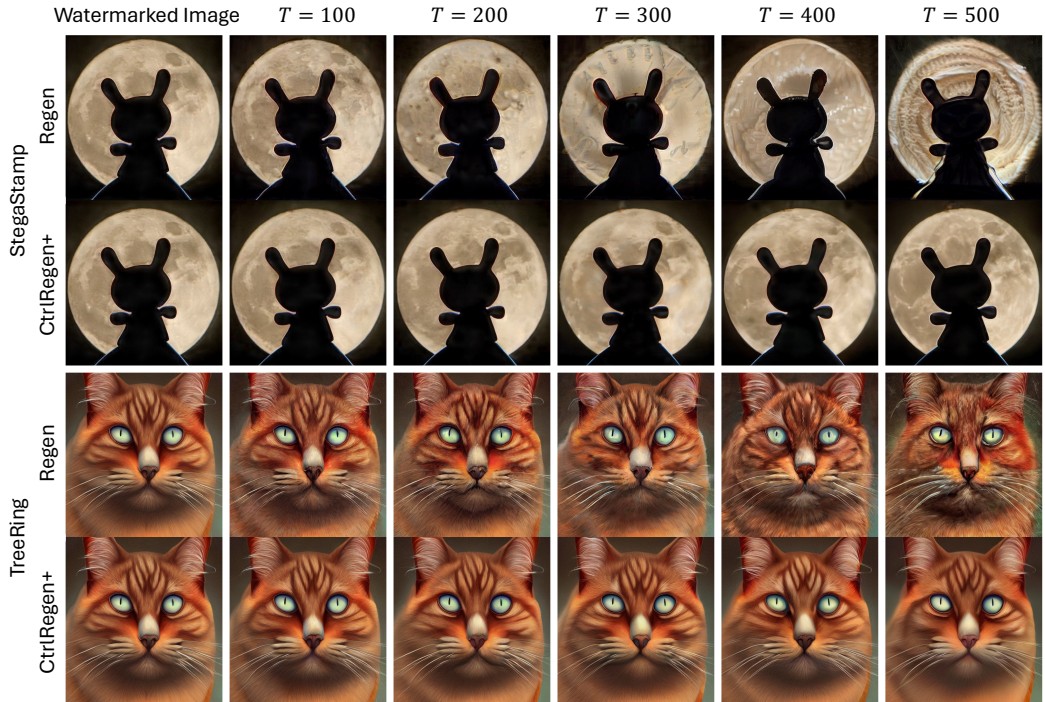

Figure 7: Visual examples of CtrlRegen+ compared to Regen at different numbers of noising steps are illustrated, with the noising step $T$ set to $\{100, 200, 300, 400, 500\}$. The image is watermarked using StegaStamp and TreeRing. It shows that our CtrlRegen+ preserves high visual consistency and quality, even across various noising steps.

Table 2: The visual similarity and quality performance between approaches using only semantic control versus those incorporating both semantic and spatial control. The results reveal that semantic control alone provides only coarse-grained guidance, while incorporating both semantic and spatial controls significantly enhances image consistency and quality across various watermarking methods.

| Watermarks | Attack Settings | CLIP-FID ↓ | PSNR ↑ | Q-Align ↑ | LIQE ↑ |
|---|---|---|---|---|---|
| DwtDctSvd | Semantic Only | 13.06 | 15.41 | 2.84 | 2.55 |
| | Semantic&Spatial | 8.68 | 19.13 | 3.63 | 3.76 |
| RivaGAN | Semantic Only | 9.06 | 15.43 | 2.86 | 2.55 |
| | Semantic&Spatial | 4.24 | 19.53 | 3.62 | 3.69 |
| SSL | Semantic Only | 11.36 | 15.72 | 2.38 | 1.78 |
| | Semantic&Spatial | 5.64 | 19.07 | 3.22 | 3.15 |
| StableSignature | Semantic Only | 4.04 | 14.37 | 3.09 | 2.71 |
| | Semantic&Spatial | 1.83 | 19.03 | 3.97 | 4.02 |
| StegaStamp | Semantic Only | 9.97 | 15.72 | 2.78 | 2.52 |
| | Semantic&Spatial | 5.27 | 19.10 | 3.62 | 3.77 |
| TreeRing | Semantic Only | 3.93 | 15.24 | 3.40 | 3.24 |
| | Semantic&Spatial | 1.63 | 19.32 | 4.17 | 4.34 |

**Inference time of our watermark removal methods.** To evaluate the inference time of our regeneration methods, we conducted experiments to compute the average inference time, including the backbone SD model using text as input, CtrlRegen, and CtrlRegen+ with different noising steps (200, 400, 600, 800, 1000). The results are presented in the following table. From Table 3, it is evident that our regeneration method (CtrlRegne, CtrlRegen+ 800, and CtrlRegen+ 1000) introduces a negligible time delay (less than 1 second) compared to the backbone Stable Diffusion model. Additionally, for CtrlRegen+ with noising steps smaller than 600, the inference time is even shorter than that of the backbone SD model. This demonstrates the efficiency of our approach.

Table 3: Inference time comparison of backbone SD, CtrlRegen, and CtrlRegen+ with varying noising steps for watermark removal. It highlights the efficiency of our methods.

| Setting | Backbone SD | CtrlRegen | CtrlRegen+ (200) | CtrlRegen+ (400) | CtrlRegen+ (600) | CtrlRegen+ (800) | CtrlRegen+ (1000) |
|---|---|---|---|---|---|---|---|
| Generation Time/s | 1.74 | 2.56 | 0.69 | 1.19 | 1.71 | 2.15 | 2.64 |

**Discussion about CtrlRegen Visual Similarity.** In order to further illustrate that CtrlRegen preserves the overall content and ensures the perturbation integrates well with the image, we provide a visual comparison between CtrlRegen and Regen at the same PSNR level. Our experiments were conducted on StegaStamp watermarks, with Regen using a noising step of 360, while CtrlRegen generates images starting from clean noise. Under these settings, both methods achieve comparable PSNR values. However, as shown in Figure 8, at similar PSNR levels, Regen introduces noticeable distortions to the image content, whereas our method maintains strong alignment with the watermarked image. This comparison effectively demonstrates that the relatively lower PSNR of our approach does not compromise the overall integrity of the watermarked image content.

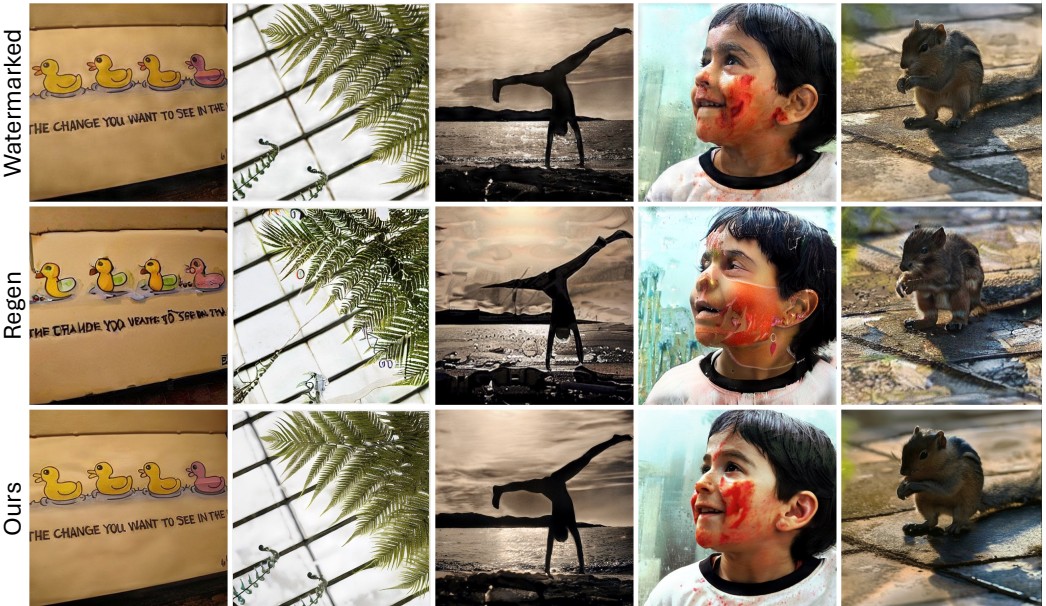

Figure 8: Visual examples of CtrlRegen compared to Regen at similar PSNR levels, where Regen uses a noising step of 360. The image is watermarked using StegaStamp. Regen introduces noticeable distortions to the image content, whereas our method maintains strong alignment with the watermarked image. This comparison effectively demonstrates that the relatively lower PSNR of our approach does not compromise the overall integrity of the watermarked image content.

Table 4: Bit number we used in our experiments for different watermarking methods.

| Watermark | DwtDctSvd | RivaGAN | SSL | StableSignature | StegaStamp |
|---|---|---|---|---|---|
| Bits | 32 | 32 | 32 | 48 | 96 |

## B    ALGORITHM OF PROPOSED METHODS

---
**Algorithm 1** CtrlRegen
---
**Input:** Watermarked image $x_w$.

 1:  $\hat{x}_w \leftarrow f(x_w)$
 2:  $z_T \leftarrow \mathcal{N}(0, I_n)$
 3:  **for** $t = T, T-1, \cdots, 1$ **do**
 4:     $\xi_t \leftarrow \epsilon_\theta(z_t, x_w, \hat{x}_w, t)$
 5:     $z_{t-1} \leftarrow z_t - \xi_t$
 6:  **end for**
 7:  $\tilde{x} \leftarrow \mathcal{D}(z_0)$

**Output:** Cleaned image $\tilde{x}$.

---

---
**Algorithm 2** CtrlRegen+
---
**Input:** Watermarked image $x_w$, Noising step $t^*$.

 1:  $\hat{x}_w \leftarrow f(x_w)$, $z_0 \leftarrow \mathcal{E}(x_w)$
 2:  $\epsilon \leftarrow \mathcal{N}(0, I_n)$, $z_{t^*} = \sqrt{\alpha_{t^*}} z_0 + \sqrt{1 - \alpha_{t^*}} \epsilon$
 3:  **for** $t = t^*, \cdots, 1$ **do**
 4:     $\xi_t \leftarrow \epsilon_\theta(z_t, x_w, \hat{x}_w, t)$
 5:     $z_{t-1} \leftarrow z_t - \xi_t$
 6:  **end for**
 7:  $\tilde{x} \leftarrow \mathcal{D}(z_0)$

**Output:** Cleaned image $\tilde{x}$.

---

