# OpenReview forum: "Image Watermarks are Removable using Controllable Regeneration from Clean Noise"
_ICLR.cc/2025/Conference — ICLR 2025 Poster_

### Official Review · Reviewer_FvFj · 2024-10-30

**Soundness:** 3
**Presentation:** 3
**Contribution:** 2
**Rating:** 6
**Confidence:** 4

**Summary:**

In this paper, the authors developed a method to remove watermarked embedded in images. They use clear noise, watermarked image embedding, and edges of watermarked images to regenerate the image, such that the image content is reminded but the watermark is removed. More precisely, they used attention mechanism to fuse the watermarked image embedding to the SD UNet and used Spatial-Net to extract edge features and add to the UNet. The authors conducted comprehensive experiments and demonstrated the proposed methods offer stronger removal effect, comparing with previous methods.

**Strengths:**

1)	The presentation is clear, and the paper is easy to follow.
2)	The use to clean noise is a good starting point
3)	The experiments demonstrate that it outperforms the other methods. The scale of the experiments is reasonable.

**Weaknesses:**

1)	Although the clean noise and edge images are unlikely contained watermark information, I feel that it is possible for it to go through the image encoder and finally, appear in the output image. The authors should discuss more why the watermark information cannot go through the semantic control network.
2)	The authors use two mechanisms to fuse semantic information and spatial information. Why don’t use the same approach, such as attention? Any reason behind of this design.
3)	The authors should revise the motivation. If the aim is to develop a stronger method to break all watermark algorithms, what is good the society and the community?
4)	I cannot find out how many bits are used in the watermarking experiments.
5)	Can the watermarking algorithms use more bits to increase the protection?
6)	Although in visual quality indexes, expect for PSNR, the proposed methods are stronger than the baselines. The images for visual comparisons are not enough. The authors should provide more visual comparison.
7)	The authors should provide a scheme to defend the proposed methods. Otherwise, this paper will cause more harm than good to the society.
8)	Do the constraints e.g., L2 norm or L infinite norm using in watermark algo development affect the removal capacity?

**Questions:**

See the weaknesses

**Details Of Ethics Concerns:**

The proposed methods can effectively remove watermarks. They may cause serious copyright issues.

---

> ### Author Response · Authors · 2024-11-21
>
> We thank the reviewer for thoroughly reading our manuscript and providing constructive feedback. We have carefully considered the reviewer’s comments and provided the following responses.
>
> **Q1. Discuss why watermark information cannot go through the semantic control network.**
>
> The intuition behind why watermark information does not propagate through the semantic control network is based on two main factors:
> High-Level Semantic Embedding: Similar to text-guided Stable Diffusion, our image-guided regeneration encodes the image into a high-level, coarse-grained semantic embedding consisting of 16 learnable tokens. The encoder significantly downsampling the image, focusing on extracting global semantic features while discarding finer details, such as pixel-level or frequency perturbations. This downsampling naturally filters out watermark information that resides in low-level details.
>
> Cross-Attention Mechanism: The cross-attention layer in the backbone U-Net aligns the latent image representation with textual or visual features to produce coherent, semantically meaningful outputs. This mechanism emphasizes semantic, structural, or stylistic aspects of the image embedding. Therefore, even if some watermark information is retained in the embedding, the cross-attention mechanism might effectively ignore it as it doesn't contribute meaningfully to the semantic structure of the output image.
>
> These two mechanisms ensure that watermark information is largely excluded from the semantic control network's influence. As mentioned by the reviewer, clean noise and edge images are unlikely to contain watermark information. The experimental results in our manuscript confirm that our attack effectively removes existing image watermarks, validating our intuition that watermark information does not propagate through the semantic control model and is absent in the regenerated image.
>
>
> **Q2. Explain the reason for using two separate mechanisms to fuse semantic and spatial information rather than a unified approach like attention.**
>
> A unified approach like attention is a good idea. However, it is challenging to use only a single image embedding to effectively capture both semantic and fine-grained spatial features simultaneously.
>
> The core mechanism behind the semantic control is attention, leveraging the inherent attention mechanism of U-Net. For semantic control, an image encoder and a projection network are employed to extract the image embedding, which consists of 16 learnable tokens. This extracted image embedding is then integrated into the U-Net through a decoupled cross-attention layer to guide the generation process. This design aligns with the intuition of the backbone Stable Diffusion model, which uses text to guide image generation via cross-attention.
>
> Initially, our design focused solely on semantic (attention-based) control. However, we observed that semantic control alone could only provide coarse-grained, semantic-level regeneration, resulting in regenerated images lacking fine-grained details. To address this limitation and improve the consistency between watermarked and regenerated images, we introduced spatial control. This mechanism incorporates the Canny edge of the image to supply fine-grained spatial information.
>
> To demonstrate the impact of semantic and spatial control, we conducted an ablation study, as shown in Table 2 of our manuscript. The results highlight that the inclusion of spatial control significantly enhances the consistency of regenerated images.
>
> **Q3. Clarify the motivation for developing a stronger method to break all watermark algorithms.**
>
> Thank you for highlighting this issue. Our primary motivation for developing a more effective watermark removal method is to uncover the vulnerabilities in current watermarking techniques. Notably, the recent NeurIPS competition on erasing invisible image watermarks seeks to assess the robustness of watermarking systems under varying visibility levels and attacker knowledge. We believe this aligns closely with our objectives. By demonstrating the weaknesses in existing watermarking methods, we aim to underscore the urgent need for more robust and resilient watermarking solutions. We will further elaborate on this motivation in the next version of our manuscript.
>
> **Q4. Clarify how many bits are used in the watermarking experiments.**
>
> Thank you for pointing out this confusion. For your convenience, we have listed the bits used for different watermarking methods in the following table. For clarity, this table will also be included in our manuscript.
>
> | Watermark | DwtDctSvd | RivaGAN | SSL | StableSignature | StegaStamp |
> |-----------|-----------|---------|-----|-----------------|------------|
> | Bits      | 32        | 32      | 32  | 48              | 96         |

---

> ### Author Response · Authors · 2024-11-21
>
> **Q5. Can the watermarking algorithms use more bits to increase the protection?**
>
> We conducted experiments to investigate the relationship between the number of bits and attack performance. For this study, we set the bits number of DwtDctSvd, RivaGAN, and SSL to 16 bits and 32 bits, respectively, and for StegaStamp, we set the bits number to 50 bits. We applied CtrlRegen+ to different watermarking methods, with the noising step set to 300 for DwtDctSvd, RivaGAN, and SSL, and 500 for StegaStamp. The results, presented in the following table, demonstrate that varying the number of bits does not enhance protection, indicating that increasing bits alone is insufficient to counteract our attack.
>
> | Watermark | DwtDctSvd-16b | DwtDctSvd-32b | RivaGAN-16b | RivaGAN-32b | SSL-16b | SSL-32b | StegaStamp-16b | StegaStamp-32b |
> |-----------|---------------|---------------|-------------|-------------|---------|---------|----------------|----------------|
> | TPR@1%FPR | 0.00          | 0.00          | 0.02        | 0.02        | 0.23    | 0.20    | 0.30           | 0.33           |
>
>
> **Q6. Provide more visual comparison.**
>
> Thank you for your valuable suggestion. For your reference, we provide more image examples (https://anonymous.4open.science/r/ICLR_13019_rebuttal-0E02/examples.pdf) across different watermarks showcasing our method on a broader range of images, including those with text and scenes.
>
> **Q7. Provide a scheme to defend the proposed methods.**
>
> Thank you for your good suggestion. As analyzed in Q1, the primary reason our watermark removal method effectively removes existing image watermarks is that current watermarking schemes are not truly semantic-based. Their watermark features are discarded during the regeneration process. To defend against our attack, one potential approach is to design a real semantic image watermark that remains intact as long as the image's semantics are preserved, allowing the watermark to survive regeneration, pass through the semantic control model, and appear in the regenerated image. This represents an intriguing research direction, and we plan to explore it further in our future work. We will also include this potential defense proposal in our manuscript.
>
> **Q8. Do the constraints e.g., L2 norm or L infinite norm used in watermark algorithm development affect the removal capacity?**
>
> We conducted experiments on the SSL watermark by reducing the $L_2$ norm constraints by lowering its weight $\lambda$, which increases distortion. We set $\lambda$ to 0.5 and 1, respectively, to observe the differences. CtrlRegen+ with 300 noising steps was applied to the watermarked images. The results, presented in the following table, suggest that the $L_2$ norm constraints of the watermarking algorithm do not affect the removal capability of our method.
>
> | Weight    | 0.5  | 1.0  |
> |-----------|------|------|
> | TPR@1%FPR | 0.22 | 0.20 |

---

> ### Author Response · Authors · 2024-11-25
> **Follow-up discussion**
>
> Dear reviewer FvFj,
>
> We sincerely appreciate your time and effort in reviewing our submission and providing valuable suggestions. While we hope to have addressed your concerns adequately, we understand there may still be areas requiring further clarification or discussion. We are fully prepared to address your outstanding issues. Should our responses have successfully addressed all your questions, we would be deeply grateful if you could consider enhancing the score to further support our submission. Thank you very much for your thoughtful review.
>
> Best Regards,
> Paper13019 Authors

---

> ### Comment · Reviewer_FvFj · 2024-11-25
>
> Dear Authors,
>
> Thank you very much for your response. You have answered all my questions. I added one score for your effort.
>
> Regards,
> Reviewer FvFj,

---

> ### Author Response · Authors · 2024-11-25
> **Thank you for the enhanced rating and your valuable suggestions**
>
> We sincerely appreciate your insightful suggestions and the upgraded score. We will integrate your valuable advice into our finalized manuscript. Thank you for your further support!
>
> Best Regards,
>
> Paper13019 Authors

---

### Official Review · Reviewer_JmKA · 2024-11-01

**Soundness:** 3
**Presentation:** 3
**Contribution:** 2
**Rating:** 8
**Confidence:** 4

**Summary:**

This paper introduces a watermark attack method capable of removing watermark embeddings through either low or high perturbations by regenerating the original image from clean gaussian noise. The approach preserves the original content by injecting features from the watermarked image into the regeneration process.

**Strengths:**

* This approach is novel; it begins with Gaussian noise and leverages a controllable removal process that injects the features of the watermarked image during the denoising process to achieve a clean, watermark-free image.

* The experimental results effectively demonstrate this method’s capability in removing watermarks across various watermarking techniques.

**Weaknesses:**

* The paper lacks an ablation study for each of the modules in the proposed method, specifically the semantic control and spatial control modules. The contribution of each control module to overall performance should be investigated.

* As shown in Table 1, the quality of the resulting images, measured by reference-based metrics, is low, indicating that the method significantly alters the images.

**Questions:**

Please refer to the Weaknesses section above.

---

> ### Author Response · Authors · 2024-11-21
>
> We thank the reviewer for thoroughly reading our manuscript and providing constructive feedback. We have carefully considered the reviewer’s comments and provided the following responses.
>
> **Q1. Lack of ablation study for each of the modules in the proposed method, specifically the semantic control and spatial control modules.**
>
> We conduct a detailed ablation study in Table 2 of Appendix A to investigate the impact of semantic and spatial control on consistency. The semantic control model extracts the semantics from the watermarked image, ensuring the basic semantic integrity of the regenerated image, though it may lack detailed accuracy. The spatial control network, on the other hand, manages more detailed aspects of the regeneration process. To demonstrate this, we regenerate the watermarked image using only semantic control and then again using both semantic and spatial controls together, allowing us to observe the enhancements in visual similarity and quality. Table 2 displays the corresponding results. It demonstrates that the usage of spatial control not only enhances the consistency of the image but also improves its quality.
>
>
> **Q2. Explain the reason why Table 1 shows low reference-based quality metrics.**
>
> Table 1 highlights the attack performance of our proposed CtrlRegen method, a robust regeneration approach that starts from clean Gaussian noise. CtrlRegen is particularly effective for high-perturbation watermarks, achieving superior watermark removal performance, where baseline methods like Regen and Rinse fail. **While CtrlRegen exhibits a relatively lower PSNR compared to these baselines, this does not imply significant image degradation or alteration caused by our method.**
>
> PSNR is a pixel-level similarity metric that quantifies the differences between corresponding pixels of two images. Since CtrlRegen regenerates images from scratch using semantic and spatial information extracted from the watermarked image, it inherently introduces high pixel-level perturbations. However, these perturbations are content-adaptive and harmonize with the semantic structure of the image without introducing unnatural distortions. As illustrated in Figure 4 of our manuscript, the residual between the regenerated and watermarked images indicates that the perturbations align well with the original image content, maintaining visual coherence. In contrast, although Regen and Rinse achieve higher PSNR scores, they often produce visible artifacts and distortions in the regenerated images, as also shown in Figure 4. Moreover, it is worth noting that, beyond PSNR, CtrlRegen outperforms the baseline methods in terms of CLIP-FID and image quality metrics such as Q-Align and LIQE, further supporting our claims.
>
> Additionally, Table 1 provides an inherently unfair comparison, as the watermarking removal performance of our method is much better than these baselines. Figures 5 and 6 in our manuscript present a more fair and comprehensive comparison by evaluating our method and the uncontrolled regeneration methods at the same level of attack performance by varying noising steps. These figures demonstrate that, at the same level of attack performance, CtrlRegen achieves better consistency and image quality (including PSNR) compared to the baselines. This highlights that our method offers a superior trade-off between attack performance and image quality.

---

> ### Author Response · Authors · 2024-11-25
> **Follow-up discussion**
>
> Dear reviewer JmKA,
>
> We sincerely appreciate your time and effort in reviewing our submission and providing valuable suggestions. While we hope to have addressed your concerns adequately, we understand there may still be areas requiring further clarification or discussion. We are fully prepared to address your outstanding issues. Should our responses have successfully addressed all your questions, we would be deeply grateful if you could consider enhancing the score to further support our submission. Thank you very much for your thoughtful review.
>
> Best Regards,
> Paper13019 Authors

---

> ### Author Response · Authors · 2024-11-25
>
> We sincerely appreciate your feedback, which has afforded us an opportunity for further discussion. We are pleased to hear that your first question is well-addressed. For your second question, we present a further response concerning the proposed problems.
>
> We want to emphasize that **our method will not alter the overall content of the watermarked image.** This is achieved through the implementation of our two control modules, particularly the spatial control module, which leverages extracted Canny edges to ensure that the regenerated image content remains confined to those edges.
>
> To further illustrate that our method preserves the overall content and ensures the perturbation integrates well with the image, we provide a visual comparison between CtrlRegen and Regen at the same PSNR level. Our experiments were conducted on StegaStamp watermarks, with Regen using a noising step of 360, while CtrlRegen generates images starting from clean noise. Under these settings, both methods achieve comparable PSNR values. However, as shown in the figure (https://anonymous.4open.science/r/ICLR_13019_rebuttal-0E02/comparison_example.pdf), at similar PSNR levels, Regen introduces noticeable distortions to the image content, whereas our method maintains strong alignment with the watermarked image. This comparison effectively demonstrates that the relatively lower PSNR of our approach does not compromise the overall integrity of the watermarked image content.
>
> We completely understand the reviewer’s concerns regarding image quality, and we hope our response helps address these concerns more effectively. We appreciate the reviewer for highlighting this concern, as it provides valuable insights that will help us further enhance our manuscript. Additionally, we will consider other metrics, such as LPIPS or SSIM, in our future works. Please don’t hesitate to let us know if the reviewer has any further questions or suggestions. Once again, we sincerely thank the reviewer for the thoughtful and constructive feedback.

---

### Official Review · Reviewer_EJxb · 2024-11-02

**Soundness:** 3
**Presentation:** 3
**Contribution:** 2
**Rating:** 6
**Confidence:** 5

**Summary:**

This paper introduces CtrlRegen, a watermark removal method that leverages controllable image regeneration from clean Gaussian noise to effectively remove state-of-the-art image watermarks, even those with high perturbation.
CtrlRegen employs a trained semantic control adapter and spatial control network to guide the denoising process using extracted semantic and spatial features from the watermarked image.
CtrlRegen+, further enhances the method by adding adjustable noise steps to the latent representation before denoising, allowing for a smoother trade-off between watermark removal and image quality.
Experiments across various watermarking techniques demonstrate CtrlRegen's superiority in visual consistency and watermark removal performance compared to existing approaches , particularly for high-perturbation watermarks like StegaStamp and TreeRing.

**Strengths:**

1. CtrlRegen uses clean Gaussian noise as a starting point for denoising in a diffusion model, while existing regeneration attacks that add limited noise to the watermarked image's latent representation
2. CtrlRegen+ adds an adjustable noising step before denoising, allowing for a trade-off between watermark removal and image quality.
3. Experiments show CtrlRegen outperforms existing regeneration methods (Regen and Rinse) in removing high-perturbation watermarks while maintaining better visual quality.
4. The method is a "no-box" attack, requiring only a watermarked image without knowledge of the watermarking scheme.  Ablation studies confirm the importance of both semantic and spatial control in maintaining image consistency.

**Weaknesses:**

1. Though the method proposes a novel approach to remove the image watermark, the watermark's perturbation is vulnerable to sophisticated attacks such as VAE. Applying VAE can also easily remove the watermark. The difficulty of watermark removal is relatively less challenging, the main contribution lies in the semantic consistency part.
2. The proposed method relies on regenerating a new clean from noise, while when there may exist large-scale watermarked image in 2K or 4K, marking the proposed method hard to regenerate a large-scale clean image while still keeping high semantic consistency.
3. The selected baselines ReGen contains theoretical explanations on why the watermarks are provably removable. The proposed method lacks a theoretical analysis of why the regeneration scheme from noise can also be proved to remove the watermark.

**Questions:**

1. What is the model size and inferencing time for the proposed method CtrlRegen and CtrlRegen+?
2. Compared with the common sophisticated attack on the watermarked image such as VAE or the recently proposed watermark removal schemes in Regen, what is the key advantage of the proposed method?

---

> ### Author Response · Authors · 2024-11-21
>
> We thank the reviewer for thoroughly reading our manuscript and providing constructive feedback. We have carefully considered the reviewer’s comments and provided the following responses.
>
> **Q1. The watermark's perturbation is vulnerable to sophisticated attacks like VAE, which can easily remove the watermark, making its removal relatively less challenging.**
>
> The VAE attack is an effective method for removing watermarks. However, it works only on low-perturbation watermarks, such as DctDwtSvd, RivaGAN, and SSL, and is ineffective against high-perturbation watermarks, such as StegaStamp and TreeRing. To demonstrate this, we apply the VAE attack to both low- and high-perturbation watermarks to evaluate its attack performance. The results are presented in the following table. The detection performance (measured by TPR@1%FPR) for low-perturbation watermarks drops below 0.5, but the VAE attack has minimal effect on high-perturbation watermarks. In contrast, our proposed controllable regeneration method effectively removes both low- and high-perturbation watermarks. Therefore, existing regeneration attacks struggle to effectively remove high-perturbation watermarks, whereas our proposed method overcomes this challenge, highlighting its significance and contribution.
>
>
> | Watermarks | DctDwtSvd | RivaGAN | SSL  | Stable Signature | StegaStamp | TreeRing |
> |------------|-----------|---------|------|------------------|------------|----------|
> | VAE        | 0.40      | 0.07    | 0.43 | 0.25             | 1.00       | 1.00     |
> | Regen      | 0.39      | 0.07    | 0.39 | 0.02             | 0.99       | 0.87     |
> | CtrlRegen  | 0.00      | 0.00    | 0.06 | 0.02             | 0.01       | 0.12     |
>
>
>
> **Q2. The proposed method faces challenges in maintaining high semantic consistency when applied to large-scale watermarked images in 2K or 4K resolutions.**
>
> Considering that the backbone diffusion model used for the regeneration task is Stable Diffusion 1.5, which supports only 512 $\times$ 512 resolutions, it is true that handling high-resolution images presents challenges. However, there are two methods to address this issue. First, since our approach does not modify the backbone, we can retrain the semantic and spatial control networks based on a diffusion model that supports higher-resolution outputs, such as Stable Diffusion XL. Second, for extremely high-resolution images, we can divide the images into smaller patches, process each patch independently, and then reconstruct the full image. It is an interesting direction, and we will explore high-resolution watermarking image removal in our future work.

---

> ### Author Response · Authors · 2024-11-21
>
> **Q3.  The proposed method lacks a theoretical analysis of why the regeneration scheme from noise can also be proved to remove the watermark.**
>
> Theoretically, by applying a similar proof technique as in Regen [1], we can establish a stronger guarantee for watermark removal. Given a watermarked image $x_w$ and an unwatermarked image $x_0$, [1] assumes that the VAE ($\phi$) of the Stable Diffusion model satisfies the Lipschitz property:
>
>   $||\phi(x_w)-\phi(x_0)|| \leq L||x_w-x_0||$.
>
> [1] further considers two Gaussian distributions, $P \sim N(\phi(x_0), \sigma^2 I)$ and $Q \sim N(\phi(x_w), \sigma^2 I)$. Let $T(P, Q): [0,1] \rightarrow [0,1]$ be the tradeoff function that outputs the Type II error of the likelihood ratio test for this binary hypothesis testing problem as a function of the Type I error. Based on Lemma 2.9 in [2], for any post-processing function $h$, $T(h(P), h(Q)) \geq T(P, Q)$. In particular, the post-processing of regeneration $h$ includes the noising process $F(\phi(x_w), t)$ and the denoising process $R(F(\phi(x_w), t))$. The watermark removal guarantee in [1] is proven by considering only the noising process, which is consistent with our method (CtrlRegen+). Thus, Theorem 4.3 in [1] is also applicable to our algorithm, i.e.,
>
> $f\left(\epsilon_1,t\right)=\Phi\left(\Phi^{-1}\left(1-\epsilon_1\right)-\frac{L||x_w-x_0||}{\sqrt{\left(1-\alpha\left(t\right)\right) / \alpha\left(t\right)}}\right)$,
>
> where $\Phi$ is the cumulative density function (CDF) of the standard normal distribution and $\epsilon_1$ represents the Type I error.
>
> However, our controllable regeneration method differs from the uncontrolled regeneration method in [1] in two key aspects: (1) **Stronger Noising Process**: Our noising process, $F(\phi(x_w), t^*)$, allows more noise to be added to $\phi(x_w)$ to destroy the watermark information thoroughly, while [1] is limited to adding only a small amount of noise to balance image consistency/quality. (2) **Controllable Denoising Process**: Our denoising process is controllable, $R_c(F(\phi(x_w), t, E(x_w)))$, leveraging high-level semantic information and fine-grained spatial information ($E(x_w)$). In contrast, [1] uses an uncontrolled denoising process. **The key advantage of our method lies in its controllable process, which ensures high image quality and consistency. This allows for the application of a stronger noising process (larger $t$), enabling more effective removal of watermark information.**
>
> For Theorem 4.3, $t \in [0,1]$ corresponds to noising steps from $0$ to $1000$ in our manuscript. As mentioned in [1] (Remark 4.4), to guarantee the quality of the regenerated image, the noising step $t'$ must be below a threshold ($t' \leq \eta$). In contrast, our controllable denoising process can select any $t^*$ within $[0,1]$ without sacrificing image quality or consistency. For instance, when $t=1$, $f(\epsilon_1)=1-\epsilon_2$, indicating a perfectly watermark-free result. However, in [1], such a high level of noise would cause the regenerated image to lose the semantic information of $x_w$ due to the uncontrolled denoising process. In contrast, our controllable regeneration process, $R_c(F(\phi(x_w), t, E(x_w)))$, reconstructs the image with high consistency and quality by utilizing its semantic and spatial information effectively. **Therefore, we can always have a $t^\*\geq t$ such that $\epsilon_2 \geq f(\epsilon_1, t^\*) \geq f(\epsilon_1, t)$, which provides a stronger watermark removal guarantee with high image consistency/quality.**
>
> Figures 5 and 6 in our manuscript support our statement. In these experiments, we set different values of $t$ for both Regen and CtrlRegen+. The results clearly demonstrate that, at the same attack performance, our method consistently achieves better image quality and consistency compared to Regen. Conversely, for the same level of image quality, our method delivers superior attack performance. This highlights the advantages of our controllable regeneration methods compared to the existing uncontrolled ones.
>
> [1] Xuandong Zhao, Kexun Zhang, Zihao Su, Saastha Vasan, Ilya Grishchenko, Christopher Kruegel, Giovanni Vigna, Yu-Xiang Wang, and Lei Li. Invisible image watermarks are provably removable using generative ai. arXiv preprint arXiv:2306.01953, 2023a.
>
> [2] Jinshuo Dong, Aaron Roth, and Weijie J Su. Gaussian differential privacy. Journal of the Royal Statistical Society Series B: Statistical Methodology, 84(1):3–37, 2022.

---

> > ### Comment · Reviewer_EJxb · 2024-11-25
> >
> > I still have some questions:
> > 1. The theorem works for the proposed CtrlRegen+ that maps watermarked image $x_{w}$ into latent space $\phi(x_{w})$. However, the proposed CtrlRegen reconstructs a clean image from clean noise. Any theorem that can explain the reconstruction from clean noise are also provable to remove the watermark?
> > 2. The proposed CtrlRegen+ can allow larger t. If we remove the semantic embedding net and spatial control net, it is basically similar to Regen. How many steps are allowed to add noise for CtrlGen and CtrlGen+ if we don't use the semantic net and spatial net?

---

> ### Author Response · Authors · 2024-11-21
>
> **Q4. What is the model size and inference time for the proposed methods, CtrlRegen and CtrlRegen+?**
>
> The trained semantic control model consists of approximately 50 million parameters, while the trained spatial control model comprises about 360 million parameters.
>
> To evaluate the inference time of our regeneration methods, we conducted experiments to compute the average inference time, including the backbone Stable Diffusion model using text as input, CtrlRegen, and CtrlRegen+ with different noising steps (200, 400, 600, 800, 1000). The results are presented in the following table. From the table, it is evident that our regeneration method (CtrlRegne, CtrlRegen+ 800, and CtrlRegen+ 1000) introduces a negligible time delay (less than 1 second) compared to the backbone Stable Diffusion model. Additionally, for CtrlRegen+ with noising steps smaller than 600, the inference time is even shorter than that of the backbone Stable Diffusion model. This demonstrates the efficiency of our approach.
>
> | Setting           | Backbone SD | CtrlRegen | CtrlRegen+ 200 | CtrlRegen+ 400 | CtrlRegen+ 600 | CtrlRegen+ 800 | CtrlRegen+ 1000 |
> |-------------------|-------------|-----------|----------------|----------------|----------------|----------------|-----------------|
> | Generation Time/s | 1.74        | 2.56      | 0.69           | 1.19           | 1.71           | 2.15           | 2.64            |
>
> **Q5. What is the key advantage of the proposed method compared to existing watermark removal attacks such as VAE or Regen?**
>
> The advantages of our proposed methods compared to existing regeneration attacks lie primarily in two aspects. For high-perturbation watermarks, our proposed methods achieve high attack performance while preserving image quality, whereas existing regeneration attacks (VAE or Regen) are ineffective at removing high-perturbation watermarks. For low-perturbation watermarks, as shown in Figures 5 and 6 in our manuscript, by varying the number of noising steps, our methods achieve better image quality at the same attack performance.

---

> ### Author Response · Authors · 2024-11-25
> **Follow-up discussion**
>
> Dear reviewer EJxb,
>
>
> We sincerely appreciate your time and effort in reviewing our submission and providing valuable suggestions. While we hope to have addressed your concerns adequately, we understand there may still be areas requiring further clarification or discussion. We are fully prepared to address your outstanding issues. Should our responses have successfully addressed all your questions, we would be deeply grateful if you could consider enhancing the score to further support our submission. Thank you very much for your thoughtful review.
>
>
> Best Regards,
>
> Paper13019 Authors

---

> ### Author Response · Authors · 2024-11-25
>
> We sincerely appreciate your feedback, which has afforded us an opportunity for further discussion. We are pleased to present the following response concerning the proposed problems.
>
> **Q1. Any theorem that can explain the reconstruction from clean noise are also provable to remove the watermark?**
>
> As described in our response to Q3, our CtrlRegen+ allows the time step $t$ to vary from 0 to 1. When $t=1$, the corresponding latent representation converges to clean noise, which aligns with the behavior of our CtrlRegen that generates directly from clean noise. Consequently, the theorem we propose also applies to the CtrlRegen method.
>
> **Q2. How many steps are allowed to add noise for CtrlGen and CtrlGen+ if we don't use the semantic net and spatial net?**
>
> If we omit both the semantic control and spatial control modules, our method is equivalent to Regen. For CtrlRegen, generating directly from clean noise without any control modules results in a completely semantically irrelevant image, losing all consistency with the watermarked image. In the case of CtrlRegen+, where no control modules are used, the parameter $t$ can be chosen between $0$ and $1$. However, as $t$ increases, the regenerated image exhibits more distortion and greater quality degradation. To illustrate this, we conducted experiments on Regen using various $t$ values ${0.02,0.1,0.2,0.3}$. The visual results are presented in the figure (https://anonymous.4open.science/r/ICLR_13019_rebuttal-0E02/Regen_different_step.pdf). From the figure, it is evident that when $t>0.1$, the regenerated image begins to display noticeable artifacts and distortions.
>
> We greatly appreciate your constructive feedback, which has been invaluable in helping us further improve our manuscript. We hope our response can adequately address the reviewer’s concerns. Please don't hesitate to let us know if you have any additional questions or suggestions. Once again, we extend our sincere thanks to the reviewer for the thoughtful and detailed feedback.

---

> > ### Comment · Reviewer_EJxb · 2024-11-26
> >
> > Thanks for your reply. My concerns are basically addressed.
> >
> > I think the work is interesting and I maintain a positive attitude.
> >
> > I will keep my score rating. Thank you for your efforts.

---

> > > ### Author Response · Authors · 2024-11-27
> > > **Thank you for your support and valuable suggestions**
> > >
> > > We sincerely appreciate your insightful suggestions. We will integrate your valuable comments into our finalized manuscript. Thank you once again for your valuable feedback.
> > >
> > > Best Regards,
> > >
> > > Paper13019 Authors

---

### Official Review · Reviewer_sQK5 · 2024-11-04

**Soundness:** 3
**Presentation:** 2
**Contribution:** 2
**Rating:** 6
**Confidence:** 4

**Summary:**

This paper introduces a watermark removal method called CtrlRegen, which regenerates watermarked images starting from clean Gaussian noise using a controllable diffusion model. The method employs a semantic control adapter and a spatial control network to ensure image quality and consistency during the denoising process. Additionally, authors propose an adjustable regeneration scheme to balance watermark removal performance and image fidelity. Experimental results demonstrate that this method offers superior visual consistency and enhanced watermark removal performance compared to existing regeneration approaches.

**Strengths:**

1. The idea of using a controllable diffusion model for watermark removal is conceptually interesting
2. Incorporation of semantic and spatial controls is a way to maintain image quality during the process.

**Weaknesses:**

1. In Table 1, the evaluation is only conducted against two baselines, Regen and Rinse. However, there are other existing watermark removal methods, such as adversarial attacks, editing attacks, or general-purpose methods like Unmarker[1], that are not included in the comparisons.

2. While the paper acknowledges that high-perturbation watermarks (e.g., StegaStamp and TreeRing) are more challenging to remove compared to low-perturbation watermarks, there is a lack of in-depth theoretical analysis or experimental results to explain why these watermarks are harder to remove. Providing more detailed insights into the mechanisms behind the difficulty of removing high-perturbation watermarks would strengthen the paper's argument.


3. The paper introduces both semantic and spatial controls to guide the watermark removal process, but it does not provide sufficient analysis.


[1] Kassis, Andre, and Urs Hengartner. "UnMarker: A Universal Attack on Defensive Watermarking." arXiv preprint arXiv:2405.08363 (2024).

**Questions:**

1.Why are the results for CtrlRegen+ not included in Table 1? How does its performance compare to the other methods across all watermarking techniques?


2. Meanwhile. why Figure 5 exlude base version of CtrlRegen?

---

> ### Author Response · Authors · 2024-11-21
>
> We thank the reviewer for thoroughly reading our manuscript and providing constructive feedback. We have carefully considered the reviewer’s comments and provided the following responses.
>
> **Q1. Include more existing watermark removal methods in the comparisons, such as adversarial attacks, editing attacks, or general-purpose methods like Unmarker.**
>
> As we mentioned in our paper, our watermark removal method is a no-box approach, meaning it does not require any prior knowledge of the watermark, nor access to the detector's API. Consequently, directly comparing our method with adversarial attacks would be unfair, as most current adversarial attack techniques depend on having watermark information, white-box access to the detector, or repeated black-box interactions with the detector's API.
>
> For a fair comparison with general-purpose methods like Unmarker, we intended to include its results. Unfortunately, to the best of our knowledge, the Unmarker code has not been released, making it difficult for us to reproduce its outcomes.
>
> To further assess our method, we performed editing-based attacks on three watermarking techniques: RivaGAN, StegaStamp, and TreeRing. These attacks involved applying transformations such as JPEG compression, Gaussian noise, and Gaussian blur. The results, presented in this figure (https://anonymous.4open.science/r/ICLR_13019_rebuttal-0E02/performance_with_edit_attack.pdf), indicate that the current watermarking methods demonstrate significant robustness against these traditional editing attacks.
>
> **Q2. Lack of in-depth theoretical analysis or experimental results to explain why high-perturbation watermarks are harder to remove. (Providing more detailed insights into the mechanisms behind the difficulty of removing high-perturbation watermarks )**
>
> As illustrated in Figure 1 (left) of our manuscript, a key difference between high-perturbation watermarks and low-perturbation watermarks is that the $L_2$ ​distance of high-perturbation watermarks in latent space, calculated between the watermarked ($x_w$) and unwatermarked ($x_0$) images after encoding via a VAE ($\phi$), is significantly larger compared to low-perturbation watermarks. For example, TreeRing embeds a watermark into the initial noise of latent space.
>
> The existing regeneration method (Regen) attempts to remove watermarks by introducing limited noise to $\phi(x_w)$ and subsequently denoising it using a diffusion model. While this method is effective for low-perturbation watermarks, it struggles with high-perturbation watermarks. This is primarily because the limited noise fails to sufficiently disrupt the watermark in the latent space, as demonstrated in Figure 1 (right) of our manuscript. As illustrated in Figure 5, increasing the noise added to $\phi(x_w)$ in the Regen method can gradually degrade the watermark, eventually making it resemble pure Gaussian noise. However, this approach significantly compromises image quality. **Therefore, the core challenge lies in the fact that high-perturbation watermarks induce substantial distribution deviations in the latent space compared to clean images.** Such deviations are difficult to remove with minor perturbations, such as the addition of limited noise, making them inherently more resistant to existing regeneration techniques.
>
> **Q3. Provide more analysis of introduced semantic and spatial controls that guide the watermark removal process.**
>
> Thank you for your valuable suggestion. The semantic and spatial controls provide coarse-grained and fine-grained guidance, respectively, to maintain image consistency and quality. We analyze their contributions from two aspects:
>
> Effectiveness Demonstration: We demonstrated the effectiveness of the designed semantic and spatial controls through comparative experiments, as shown in Table 1 and Figures 4, 5, and 6 in the manuscript. These experiments highlight the improvements in visual quality (measured by Q-Align and LIQE metrics) and consistency (measured by CLIP-FID and PSNR) achieved with our method compared to baseline approaches.
>
> Ablation Study: We conducted an ablation study to isolate the individual contributions of semantic control and spatial control to the overall regeneration quality, as shown in Table 2 of the manuscript. The results reveal that semantic control alone provides only coarse-grained guidance, while incorporating both semantic and spatial controls significantly enhances image consistency and quality across various watermarking methods.
>
> These findings underscore the importance of combining both semantic and spatial controls in achieving superior regeneration performance. We thank the reviewer’s suggestions, and if you have any further confusion or concerns about the semantic and spatial controls, please do not hesitate to let us know.

---

> ### Author Response · Authors · 2024-11-21
>
> **Q4. Why are the results for CtrlRegen+ not included in Table 1? How does its performance compare to the other methods across all watermarking techniques?**
>
> The watermark removal performance of CtrlRegen+ is detailed in Figure 5 and Figure 6 of our manuscript. CtrlRegen+ is not included in Table 1 because it is an adjustable attack method that allows for varying amounts of noise to be added. In Figures 5 and 6, we compare CtrlRegen+ with existing regeneration methods in terms of attack performance, consistency, and quality across all six watermarking techniques. By varying the number of noising steps, we demonstrate how CtrlRegen+ achieves a smooth trade-off between watermark removal performance and the preservation of image quality and consistency.
>
> **Q5. Why does Figure 5 exclude the base version of CtrlRegen?**
>
> The performance of CtrlRegen is included in Figure 5 for high-perturbation watermarks (StegaStamp and TreeRing) as the leftmost point of the orange line, where it has been combined into CtrlRegen+'s line. Since high-perturbation watermarks require more noise to effectively destroy the watermark information, we set the noising steps for these cases to {100, 200, 300, 400, 500, 1000}, along with the base version of CtrlRegen.
>
> For low-perturbation watermarks, we did not include the CtrlRegen baseline because we found that a low number of noising steps is sufficient to remove such watermarks. For these cases, the noising steps are set to {20, 40, 60, 80, 100, 120, 140}.
> We apologize for any confusion caused by this setup. We will clarify the experimental settings in our next manuscript to ensure better transparency and understanding.

---

> > ### Comment · Reviewer_sQK5 · 2024-11-26
> >
> > Thanks to the authors for providing more detailed experiments. Most of my concerns have been addressed.
> >
> > However, I strongly recommend that the authors provide an anchor or symbol to represent the base version of CtrlRegen to prevent confusion. Additionally, as the authors stated, the noise step settings should be clarified if the low-perturbation watermark experiments do not include the CtrlRegen baseline.
> >
> > If you incorporate above revision into your manuscript, I will align with the view that this paper meets the acceptance standards of ICLR, and I have therefore raised my score.

---

> ### Author Response · Authors · 2024-11-25
> **Follow-up discussion**
>
> Dear reviewer sQK5,
>
> We sincerely appreciate your time and effort in reviewing our submission and providing valuable suggestions. While we hope to have addressed your concerns adequately, we understand there may still be areas requiring further clarification or discussion. We are fully prepared to address your outstanding issues. Should our responses have successfully addressed all your questions, we would be deeply grateful if you could consider enhancing the score to further support our submission. Thank you very much for your thoughtful review.
>
> Best Regards,
>
> Paper13019 Authors

---

> ### Author Response · Authors · 2024-11-27
> **Thank you for the enhanced rating and your valuable suggestions**
>
> We are deeply grateful for the reviewer's feedback and insightful suggestions. We assure the reviewer that we will incorporate the reviewer's advice on revising and clarifying the experiments in the finalized version of our manuscript.  Thank you once again for your valuable feedback and support.
>
> Best Regards,
>
> Paper13019 Authors

---

### Official Review · Reviewer_6bHr · 2024-11-04

**Soundness:** 3
**Presentation:** 3
**Contribution:** 2
**Rating:** 6
**Confidence:** 4

**Summary:**

1. The paper introduces CtrlRegen, a new method for removing image watermarks using controllable diffusion models. The method starts with clean Gaussian noise and regenerates the original image, using a semantic control adapter and spatial control network to guide the denoising process.
2. The paper also introduces CtrlRegen+ which is another version of CtrlRegen that allows for adjustable watermark removal. This method  starts with noising the latent representation of a watermarked image instead.
3. Compared to existing approaches, these methods achieve better visual consistency and are able to remove robust watermarks and handle both low and high perturbation watermarks.

**Strengths:**

Originality and Significance:
The paper has novel ideas and also creatively uses existing ideas
1. Novel Idea: The method CtrlRegen is a new method for removing image watermarks using controllable diffusion models, starting from clean Gaussian noise. This enables it to handle high perturbation attacks which the previous methods ex: Regen were unable to.
2. Creative combinations of existing ideas: ControlNet like spatial control network and semantic control adapter to guide the denoising process, ensuring consistency between the original and cleaned images.
3. CtrlRegen+ enables an adjustable regeneration scheme - it allows for varying degrees of watermark destruction to counter different watermark strengths.

Clarity: The paper clearly explains the main ideas and method, provides context and presents quantitative results and comparisons in a clear manner.

Quality:
1. Results: The method outperforms previous regeneration approaches (Regen and Rinse) and demonstrates this through quantitative evaluation on both low and high perturbation watermarks. Multiple metrics are used to asses visual similarity and quality such as CLIP-FID, PSNR, Q-Align, LIQE beside evaluations related to watermark detection performance.
2. Consideration for use cases: The introduction of CtrlRegen+ shows consideration for different use cases and watermark strengths.

**Weaknesses:**

1. The paper does not outline/show any failure cases, limitations or drawbacks of the method.
2. The paper does not provide sufficient qualitative comparisons comparing the regeneration methods on diverse images  (ex: images containing text, scenes).

**Questions:**

Please see weaknesses section above

---

> ### Author Response · Authors · 2024-11-21
>
> We thank the reviewer for thoroughly reading our manuscript and providing constructive feedback. We have carefully considered the reviewer’s comments and provided the following responses.
>
> **Q1. Outline/show any failure cases, limitations, or drawbacks of the method.**
>
> Our proposed methods have two limitations.
>
> Resolution Limitation: The current semantic and spatial control networks are trained on Stable Diffusion 1.5, which is limited to generating images with a resolution of 512×512. This constraint poses challenges when handling higher-resolution watermarked images, such as those with a resolution of 1024×1024. To address this, we propose two strategies: (a) since our approach does not modify the backbone, we can retrain the semantic and spatial control networks using a diffusion model capable of higher-resolution outputs, such as Stable Diffusion XL; and (b) for extremely high-resolution images, we can divide them into smaller patches, process each patch independently, and then reconstruct the full image.
>
> Training Resource Requirements: Training the semantic and spatial control networks is both time- and resource-intensive. The process requires eight Nvidia V100 GPUs and takes approximately one week to complete. However, once the model is fully trained, the inference process is highly efficient. It requires only a single Nvidia 4090 GPU, with no significant delay compared to the original backbone.
>
>
>
> **Q2. Provide sufficient qualitative comparisons comparing the regeneration methods on diverse images (eg., images containing text, and scenes).**
>
> Thank you for your good suggestion. To further demonstrate the effectiveness of our proposed methods across a diverse range of images, we conducted experiments on the COCO-Text dataset. This dataset comprises images that depict complex scenes and contain embedded text. We consider two different watermarking techniques—StegaStamp and RivaGAN—to watermark the images. For StegaStamp, the noise levels were set at {100, 200, 300, 400, 500, 1000}. For RivaGAN, the noise levels were set at {20, 40, 60, 80, 100, 120, 140}. The results are provided in the following link (https://anonymous.4open.science/r/ICLR_13019_rebuttal-0E02/performance_on_coco.pdf). The figure demonstrates the effectiveness of our methods compared to existing baseline methods across diverse image types. Additionally, we have included more watermark removal examples for your reference (https://anonymous.4open.science/r/ICLR_13019_rebuttal-0E02/examples.pdf), showcasing our method on a broader range of images, including those with text and scenes.

---

> ### Author Response · Authors · 2024-11-25
> **Follow-up discussion**
>
> Dear reviewer 6bHr,
>
> We sincerely appreciate your time and effort in reviewing our submission and providing valuable suggestions. While we hope to have addressed your concerns adequately, we understand there may still be areas requiring further clarification or discussion. We are fully prepared to address your outstanding issues. Should our responses have successfully addressed all your questions, we would be deeply grateful if you could consider enhancing the score to further support our submission. Thank you very much for your thoughtful review.
>
> Best Regards,
>
> Paper13019 Authors

---

> ### Comment · Reviewer_6bHr · 2024-11-25
>
> Thank you to the authors for their efforts and answering the questions.
>
> 1. The 2nd question has been answered
> 2. For the first question - the failure cases part has not been addressed.
>
> I will maintain the current score.

---

> ### Author Response · Authors · 2024-11-25
>
> We sincerely appreciate your feedback, which has provided us with an opportunity for better refinement. We are glad to hear that your second question has been addressed. Regarding your first question, we will conduct more experiments on a broader range of watermarking methods to further investigate the limitations of our proposed approach by ourselves. Once again, we sincerely thank the reviewer for the thoughtful and constructive suggestions.

---

### Comment · Area_Chair_cAav · 2024-11-25

Hi Reviewers,

We are approaching the deadline for author-reviewer discussion phase. Authors has already provided their rebuttal. In case you haven't checked them, please look at them ASAP. Thanks a million for your help!

---

### Meta-Review · Area_Chair_cAav · 2024-12-19

**Metareview:**

This paper works on watermark removal. To achieve this, authors regenerates the watermarked images starting from a clean Gaussian noise via a controllable diffusion model. From experimental results, authors showed that proposal method is better than previous work.

Reviewers unanimously agreed this paper above the acceptance threshold.

Reviewers thought: 1) the ideas of this paper were novel, creative; 2) paper was well written: 3) results were good. Reviewers raised some concerns in the original feedback. But agreed that their concerns were addressed by authors during rebuttal and discussion phase. Given these, AC decided to accept this paper.

**Additional Comments On Reviewer Discussion:**

Reviewers raised concerns on 1) needs more ablation studies and experiments; 2) needs detailed insights into the mechanisms behind the difficulty of removing high-perturbation watermarks; 3) methods significantly alters the images. Authors resolved reviewers' concerns through more experimental results, theoretical analysis or detailed explanations.

---

### Decision · Program_Chairs · 2025-01-22

Accept (Poster)